# The genetic factors of bilaterian evolution

Peter Heger[1]*, Wen Zheng[1†], Anna Rottmann[1], Kristen A Panfilio[2,3], Thomas Wiehe[1]

[1]Institute for Genetics, Cologne Biocenter, University of Cologne, Cologne, Germany; [2]Institute for Zoology: Developmental Biology, Cologne Biocenter, University of Cologne, Cologne, Germany; [3]School of Life Sciences, University of Warwick, Gibbet Hill Campus, Coventry, United Kingdom

**Abstract** The Cambrian explosion was a unique animal radiation ~540 million years ago that produced the full range of body plans across bilaterians. The genetic mechanisms underlying these events are unknown, leaving a fundamental question in evolutionary biology unanswered. Using large-scale comparative genomics and advanced orthology evaluation techniques, we identified 157 bilaterian-specific genes. They include the entire Nodal pathway, a key regulator of mesoderm development and left-right axis specification; components for nervous system development, including a suite of G-protein-coupled receptors that control physiology and behaviour, the Robo-Slit midline repulsion system, and the neurotrophin signalling system; a high number of zinc finger transcription factors; and novel factors that previously escaped attention. Contradicting the current view, our study reveals that genes with bilaterian origin are robustly associated with key features in extant bilaterians, suggesting a causal relationship.

*For correspondence:
peter.heger@uni-koeln.de

Present address: †West China-Washington Mitochondria and Metabolism Research Center, West China Hospital, Sichuan University, Chengdu, China

Competing interests: The authors declare that no competing interests exist.

## Introduction

The taxon Bilateria consists of multicellular animals with bilateral body symmetry and constitutes a major and ancient radiation of animals. There is compelling morphological and molecular evidence for the monophyly of bilaterians (*Hejnol et al., 2009*; *Dunn et al., 2014*; *Cannon et al., 2016*), for their subdivision into protostomes and deuterostomes (*Aguinaldo et al., 1997*; *Philippe et al., 2005*; *Dunn et al., 2008*; *Simakov et al., 2013*; *Cannon et al., 2016*), and for the overall relationships of ~25 phyla that make up this group (*Dunn et al., 2008*; *Hejnol et al., 2009*; *Dunn et al., 2014*). In contrast, the evolutionary relationships of non-bilaterian metazoans are still a matter of debate, in particular the relative positions of placozoans, ctenophores, and sponges (*Brooke and Holland, 2003*; *Ryan et al., 2013*; *Pisani et al., 2015*; *Feuda et al., 2017*; *Simion et al., 2017*; *Whelan et al., 2017*).

The first unambiguously bilaterian fossils appear in Cambrian sediments with an age of ~540 million years (*Marshall, 2006*; *Erwin and Valentine, 2013*). By the end of Cambrian stage 3 (499 Mya), stem groups of all major bilaterian phyla inhabited Earth. This abrupt appearance of most bilaterian body plans, the sets of morphological features common to a phylum, already puzzled Darwin (*Darwin, 2009*). It is considered one of the most important evolutionary events after the origin of life (*Conway Morris, 2006*; *Budd, 2008*) and still awaits an explanation today. Importantly, no new body plans evolved in the 500 My since the initial radiation.

Abiotic, ecological, and genetic factors have been proposed to explain the Cambrian radiation. While deep-ocean oxygenation (*Canfield et al., 2007*), the availability of calcium (*Jackson et al., 2010*), or ecological interactions (*Budd and Jensen, 2017*) likely played a role, genetic changes in the bilaterian ancestor must ultimately have constituted its molecular basis. However, evidence for such genetic changes is scarce. Genomic sequencing of non-bilaterian animals revealed that the major signalling pathways and many developmentally important genes of bilaterians are also present in non-bilaterians, indicating that these genes evolved before the advent of bilaterians

(*Technau et al., 2005*; *Putnam et al., 2007*; *Srivastava et al., 2008*; *Srivastava et al., 2010*; *Ryan et al., 2013*; *Babonis and Martindale, 2017*). Similarly, epigenetic mechanisms to regulate gene expression, such as DNA methylation and histone modifications, seem to be conserved between bilaterians and non-bilaterian metazoans (*Zemach et al., 2010*; *Schwaiger et al., 2014*). Therefore, the common view is that modification of existing gene regulatory networks rather than the invention of new genes determined the evolution of complex body plans (*Davidson and Erwin, 2006*; *Su and Yu, 2017*).

Nevertheless, a number of studies identified genes that emerged in the ancestor of bilaterians. One example is a major expansion of miRNA families that likely triggered an increase in miRNA-mediated gene regulation (*Prochnik et al., 2007*; *Wheeler et al., 2009*). However, the significance of this event at the base of the Bilateria is unclear because frequent miRNA expansions are seen in various lineages over time (*Peterson et al., 2009*). Similarly, a link between the genome organiser CTCF and Hox genes presumably emerged in the bilaterian ancestor and might have contributed to the organisation of bilaterian body plans (*Heger et al., 2012*). The importance of CTCF for Hox gene expression has been shown repeatedly (*Mohan et al., 2007*; *Kim et al., 2011*; *Rousseau et al., 2014*; *Narendra et al., 2015*), yet direct evidence for the involvement of a Hox-CTCF link in body patterning is lacking. Another study implicated the TATA-box-binding protein-related factor 2 (TRF2) in the evolution of bilaterians. This factor may have founded new, TATA box-independent transcriptional programs involved in body plan development (*Duttke et al., 2014*), but the consequences of this hypothesis have not been tested.

Therefore, a comprehensive screen for bilaterian-specific genes and an assessment of their evolutionary impact is missing. A major obstacle for such a screen is the uneven coverage of the animal tree with sequence data. While some lineages, particularly those including model organisms (*e.g.*, nematodes, flies, or mammals), are well represented, other areas of the metazoan tree are remarkably under-represented, for example lophotrochozoans and non-bilaterian metazoans. For instance, the leading orthology databases OrthoDB (*Kriventseva et al., 2015*; *Kriventseva et al., 2019*), eggNOG (*Huerta-Cepas et al., 2016*), and OrthoMCL (*Li et al., 2003*) each contain fewer than ten non-bilaterian species, and two of these databases do not contain lophotrochozoans at all (*Figure 1*, *Table 1*). It is therefore difficult to deduce from such databases the genes that are widespread in bilaterians and absent in non-bilaterians. In addition to the bias in coverage, sequence databases suffer from annotation errors, which particularly affect non-model organisms and under-represented parts of the tree, such as non-bilaterian metazoans and lophotrochozoans. Annotation errors, in turn, have been found as the largest single source for errors in orthology benchmark testing and, together with uneven phylogenetic coverage, accounted for up to 40% of incorrect assignments (*Trachana et al., 2011*).

To address these biases and to infer bilaterian-specific genes in a reliable and robust way, we (i) assembled a dataset covering the animal tree in the most comprehensive and representative way so far; (ii) particularly strengthened resolution at the base of the Bilateria; (iii) reduced annotation errors by incorporating newly generated ORF (open reading frame) data sets; and (iv) evaluated the composition of the generated orthologous groups in a phylogenetic context. Using this strategy we extracted, from an initial set of 124 million sequences from 273 species, 157 high-confidence bilaterian-specific genes, with many functions connected to key bilaterian features.

## Results

### Dataset generation and orthogroup evaluation

Non-bilaterian metazoans are severely under-represented in existing sequence collections, but sufficient coverage is critical to illuminate bilaterian evolution. To maximise phylogenetic resolution at the origin of Bilateria, we assembled a new database specifically tailored to this purpose, the Big-WenDB (*Figure 1*, *Figure 1—figure supplement 1*; *Table 1*). This database combines sequence data of 273 species from three sources. The backbone of our analysis is the opisthokont sequence space (primarily fungi, vertebrates, and insects): 204 species, each with >8000 available sequences at GenBank, totalling 2.7 million sequences (*Table 2*; NCBI GenBank release 203 from 15 August 2014). The second part derives from transcriptome sequences of 64 species from various sources (*Supplementary file 1*–Supplementary Table 1, *Supplementary file 1*–Supplementary Table 2,

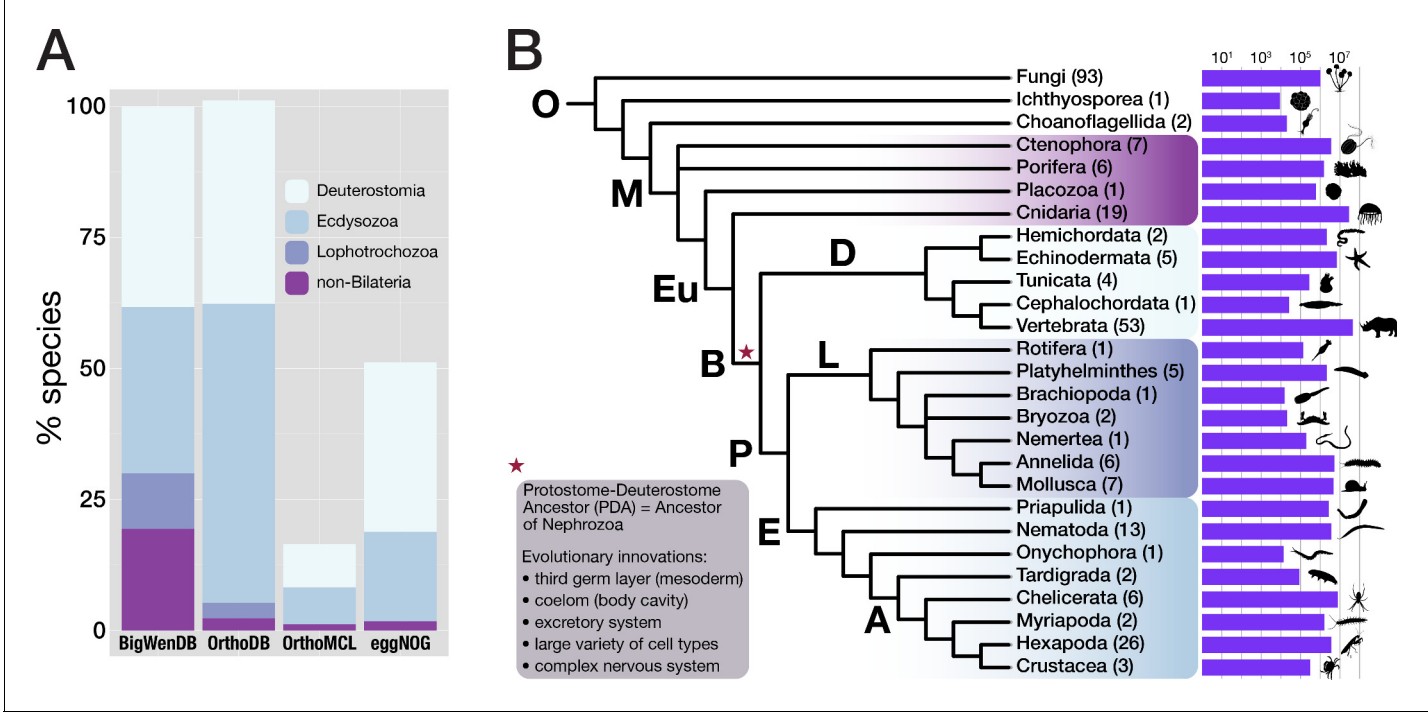

**Figure 1.** Properties of the BigWenDB data collection. (**A**) Comparison of three major orthology databases with the BigWenDB. The relative contribution of four metazoan clades (Deuterostomia, Ecdysozoa, Lophotrochozoa, and the paraphyletic group "non-Bilateria") is shown as stacked bar graph. The count of metazoans in our database (175 species) is set to 100%. In comparison to other databases, the BigWenDB has a larger repertoire of critical lophotrochozoans and non-bilaterian Metazoa. (**B**) Consensus phylogeny describing the relationships of 21 metazoan phyla covered in our database, after *Laumer et al., 2015*; *Telford et al., 2015*; *Torruella et al., 2015*; *Cannon et al., 2016*. Bold labels to the left or above branches indicate its ancestor (A: Arthropoda, B: Bilateria, D: Deuterostomia, E: Ecdysozoa, Eu: Eumetazoa, L: Lophotrochozoa, M: Metazoa, O: Opisthokonta, P: Protostomia). Numbers in parentheses (after the phylum name) indicate the number of species present from this phylum. Horizontal bars visualise the number of database sequences that belong to a given phylum (logarithmic scale; transcriptomic, ORF, and NCBI sequences summed up). Species silhouettes were downloaded from www.phylopic.org. Morphological innovations of Bilateria according to *Baguñà et al., 2008* are highlighted in a shaded box.

The online version of this article includes the following figure supplement(s) for figure 1:

**Figure supplement 1.** Phylogenetic distribution of the BigWenDB.

**Figure supplement 2.** Size distribution of three sequence data types present in the BigWenDB.

**Figure supplement 3.** ORF size distribution for 25 species with genomic data.

*Supplementary file 2*). Among others, non-bilaterian metazoans (30 species) and lophotrochozoans (12 species) contribute 11.7 million sequences to this group, complementing their poor GenBank representation (*Figure 1—figure supplement 1*). The third and largest sequence set contains ~109 million open reading frames (ORFs) obtained by translating 25 metazoan genomes (*Supplementary file 1*–Supplementary Table 3). All non-bilaterian and lophotrochozoan whole genome sequences available at the time, as well as genomes from additional phyla, were included to compile a comprehensive and representative dataset (*Figure 1—figure supplement 1*). As this strategy caused a large increase in sequence number, we limited the third set to 25 species to maintain technical feasibility. The final dataset combines 124 million sequences from 21 metazoan and three outgroup phyla, including several taxa absent from other databases, for example tardigrades, a priapulid, bryozoans, a nemertean, a rotifer, a brachiopod, and choanoflagellates (*Figure 1*, *Figure 1—figure supplement 1*).

To be able to generate clusters of orthologous proteins from this large dataset, we adapted the OrthoMCL pipeline (*Li et al., 2003*) and improved its scalability (see Appendix 1: Orthology pipeline and clustering; *Supplementary file 1*–Supplementary Table 4). As a large proportion of the resulting 824,605 orthogroups was small and had phylogenetically inconsistent composition (*Appendix 1—figure 1*; *Supplementary file 1*–Supplementary Table 5), we focused our analysis on 75,744

**Table 1.** Comparison of three major orthology databases with the BigWenDB.

The number of species of a given taxon (left column) in four different orthology databases is shown. In contrast to other databases, the BigWenDB has substantially more sequence information from non-bilaterian metazoans and therefore a better resolution at the divergence of bilaterians and non-bilaterians. D = Deuterostomia, E = Ecdysozoa. Note the bias of other databases towards insects and vertebrates, which continues in the latest database versions (*e.g.*, OrthoDB v10.2; *Kriventseva et al., 2019*).

| Taxon | OrthoDB V8 | eggNOG V4.5 | OrthoMCL V5 | BigWenDB |
|---|---|---|---|---|
| Cellular organisms | 3,027 | 2,031 | 150 | 273 |
| Metazoa | 173 | 88 | 29 | 175 |
| Bilateria | 169 | 85 | 27 | 142 |
| non-Bilateria | 4 | 3 | 2 | 33 |
| Ecdysozoa (E) | 97 | 29 | 12 | 54 |
| E w/o insects | 17 | 9 | 4 | 29 |
| Lophotrochozoa | 5 | 0 | 0 | 18 |
| Deuterostomia (D) | 66 | 55 | 14 | 65 |
| D w/o vertebrates | 5 | 4 | 1 | 12 |

orthogroups (OGs) with at least 10 species. They provide a rich repertoire for the identification of lineage-specific protein sets.

Hundreds to thousands of novel translated ORFs exist in humans and other animals, that are missed by traditional annotation methods (*Ladoukakis et al., 2011*; *Mackowiak et al., 2015*; *Raj et al., 2016*). A key aspect of our analysis is therefore the inclusion of genomic ORFs. To estimate their contribution to the clustering process, we examined the composition of all orthogroups. Genomic ORFs constitute a substantial fraction of the majority of orthogroups, comprising >90% of all sequences in 50% of orthogroups. This demonstrates that a high percentage of orthogroups is either dependent on or substantially affected by the inclusion of ORFs. Although most ORFs are short (mean length of 60 AA; *Figure 1—figure supplement 2*, *Figure 1—figure supplement 3*), nearly 2.3 million ORFs (on average 90,443 per species) are >132 AA, the mean size of domains in the PFAM database, ensuring the possibility of annotating ORF-dominated orthogroups (*Figure 1—figure supplement 2*).

**Table 2.** Composition of the BigWenDB.

The number of sequences (overall: 124,031,501) collected from three different sources (NCBI, Transcriptome, ORFs) is indicated for major taxonomic groups of the BigWenDB. "Others" comprises the ichthyosporean *Capsaspora owczarzaki* and the choanoflagellates *Monosiga brevicollis* and *Salpingoeca rosetta*.

| Group | (Super)Phylum | # Species | NCBI | Transcriptome | ORFs |
|---|---|---|---|---|---|
| Bilateria | Deuterostomia | 65 | 895,084 | 2,292,541 | 51,922,654 |
| | Ecdysozoa | 54 | 511,663 | 2,150,424 | 17,338,026 |
| | Lophotrochozoa | 23 | 170,379 | 2,618,518 | 9,805,405 |
| Non-Bilat. | Ctenophora | 7 | 0 | 1,468,372 | 2,458,546 |
| | Placozoa | 1 | 11,215 | 0 | 590,820 |
| | Porifera | 6 | 8,836 | 539,299 | 1,008,535 |
| | Cnidaria | 19 | 36,873 | 2,361,032 | 26,443,358 |
| Fungi | | 93 | 1,032,299 | 0 | 0 |
| others | | 3 | 29,292 | 0 | 0 |
| total | | 273 | 2,695,641 | 11,768,516 | 109,567,344 |

We next assessed the accuracy and biological validity of our orthogroup dataset via several approaches. First, we compared our clustering results with an external benchmark set of 70 manually curated orthogroups (*Trachana et al., 2011*; see Appendix 1: Cluster evaluation and quality control; *Supplementary file 3*). We then specifically examined the clustering results of a highly conserved and difficult to assess class of proteins, the Nkx homeodomain proteins (*Supplementary file 1*–Supplementary Table 6). Third, we evaluated potential sources of error with respect to the phylogenetic composition of a given orthogroup (see Appendix 1: Identification of bilaterian-specific genes). For this purpose, we developed a new reciprocal HMM-HMM comparison step. It performs sensitive, BLAST-independent searches for orthogroups with similar sequence profiles to validate orthogroup completeness. We demonstrated the value of this step by using two proteins as test cases, the FGF signalling pathway component Sprouty and the insulator protein GAGA factor (see Appendix 1: Identification of bilaterian-specific genes; *Supplementary file 1*–Supplementary Table 7). After these quality control steps, we finally identified 157 orthogroups as a minimal set of high confidence, bilaterian-specific orthogroups (*Supplementary file 4*).

## The domain repertoire of bilaterian-specific proteins is enriched for DNA-binding

To reveal the putative function of the 157 identified bilaterian-specific genes, we first determined their protein domain repertoire and the gene ontology terms for molecular function associated with these domains. We then compared the results to analyses carried out for the vertebrate and arthropod nodes, as these nodes represent major radiations that are well-supported by genome sequence data. The obtained terms indicate that membrane processes, including cell adhesion, G-protein-coupled receptor signalling, and $Ca^{2+}$-binding, as well as protein interactions and metal ion binding, are prominent molecular functions of bilaterian-specific proteins (*Figure 2* left, top and middle row). In contrast, terms derived from the arthropod and vertebrate nodes are markedly different. While the vertebrate repertoire comprises G-protein-coupled receptors, cadherins, and extracellular domains required for protein-protein or protein-ligand interactions, arthropod-specific genes are characterised by a broad spectrum of similarly prominent functions, from expected roles in cuticle and chitin biology to a plenitude of conserved domains of unknown function (*Figure 2* middle and right, top and middle row). These results indicate that proteins with distinct functions characterise the evolution of each of the three nodes.

Further, our comparative analysis implied that a large number of transcription factors emerged in the bilaterian ancestor. While 3.58% of vertebrate-specific orthogroups and 9.30% of arthropod-specific orthogroups had transcription factor-associated domains such as zinc fingers or homeodomains, the corresponding fraction was 26.06% in bilaterian-specific orthogroups (*Figure 2* middle row). To substantiate this result, we randomly selected 10 times 157 proteins from a curated set of 20,205 human proteins. The average number of transcription factors in these control sets was $12.8 \pm 4.44$ as opposed to 37 transcription factors in the set of 157 bilaterian-specific genes. This is a highly significant result under a number of assumptions for data distribution (see Materials and methods), lending statistical support to an unexpectedly high number of transcription factors in the bilaterian-specific dataset.

Importantly, many of the transcription factors contained tandem $C_2H_2$ zinc finger domains and already originated with multiple zinc fingers, as their extant *Drosophila* and human orthologues suggest (*Supplementary file 1*–Supplementary Table 8). With the addition of at least 13 members, the modest poly-ZF repertoire at the dawn of metazoans thus almost doubled in the bilaterian ancestor (*Figure 2—figure supplement 1*) in line with previous evidence that poly-ZF proteins emerged from a small group of eukaryotic zinc finger transcription factors (*Emerson and Thomas, 2009*). Considering that several factors with this domain configuration are involved in regulating chromatin architecture, including CTCF (*Phillips-Cremins et al., 2013*), YY1 (*Weintraub et al., 2017*), Pita (*Kyrchanova et al., 2017*), SuHw (*Van Bortle et al., 2012*), and Casz1 (*Mattar et al., 2018*), these findings open the possibility that multiple poly-ZF factors participated in modifying higher-order chromatin structure during the emergence of bilaterians, as proposed for CTCF (*Heger et al., 2012*; *Vietri Rudan and Hadjur, 2015*; *Acemel et al., 2017*). With the exception of YY1 (OG_3966: metazoan origin or earlier), all known chromatin architectural proteins emerged in the ancestor of bilaterians or later (*Heger et al., 2013*; *Heger and Wiehe, 2014*), suggesting that a more sophisticated regulation of gene expression by influencing chromatin architecture contributed to bilaterian

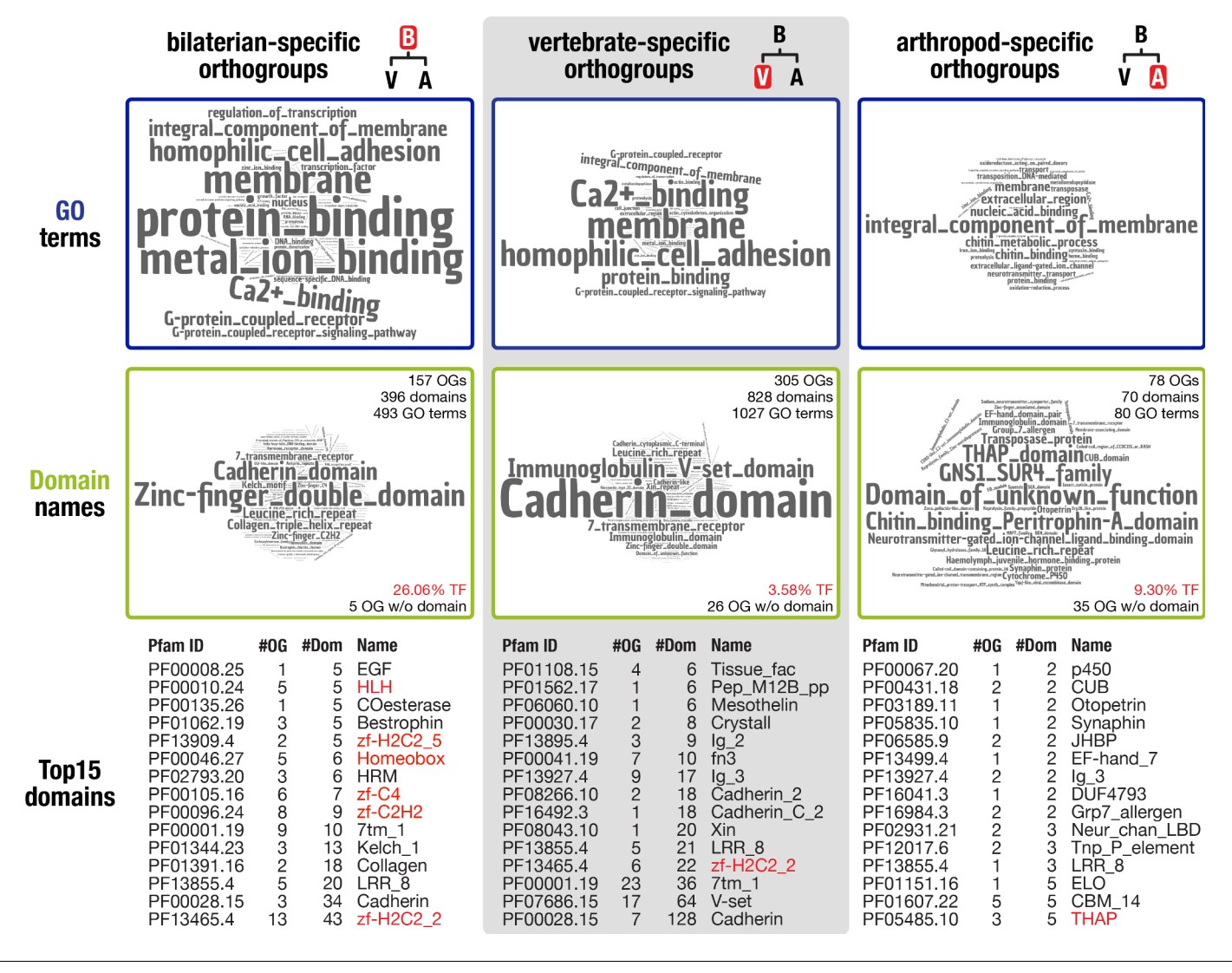

**Figure 2.** Inventory of protein domains and associated GO terms for three animal lineages.

The online version of this article includes the following figure supplement(s) for figure 2:

**Figure supplement 1.** Metazoan poly-zinc finger transcription factor repertoire and evolution.

**Figure supplement 2.** Multiple sequence alignments of two bilaterian-specific orthogroups without known domains.

**Figure supplement 3.** Multiple sequence alignments of OG_13336 and OG_31055, two bilaterian-specific orthogroups without known domains.

**Figure supplement 4.** Multiple sequence alignment of OG_8220, another bilaterian-specific orthogroup without known domains.

evolution. More generally, we note that poly-ZF proteins often comprise the most abundant transcription factor superfamily in bilaterians, with many lineage-specific expansions even within orders and families (*Panfilio et al., 2019*). Below, we also comment both on similar patterns in other protein classes and on potential other roles of a bilaterian expansion in poly-ZF proteins.

## Bilaterian-specific proteins contain novel protein domains

Using domain scans, we could not identify known protein domains or other functional annotation for 5 of the 157 bilaterian-specific orthogroups. Nevertheless, the corresponding alignments displayed extended regions of sequence conservation (*Figure 2—figure supplement 2*, *Figure 2—figure supplement 3*, *Figure 2—figure supplement 4*) arguing that these regions may constitute so far undetected protein domains. To explore whether the putative domains are bilaterian novelties, we converted them to hidden Markov models and used these to search our database of 824,605

orthogroup HMMs. In these searches, only one of the five domains showed weak evidence for homology outside the Bilateria, indicating that a protein with a similar domain exists in non-bilaterians. The other four domains were restricted to bilaterians, like the proteins they belong to (*Supplementary file 1*–Supplementary Table 9), a finding compatible with the de novo birth of these five genes. Similarly, sequences without known protein domains were also detectable in arthropod- and vertebrate-specific orthogroups (*Figure 2*) and, more generally, in approximately 40% of the 69,114 orthogroups with more than ten species. These findings open the possibility that, across opisthokonts, many lineage-specific genes are uncharacterised and may contain previously undescribed protein domains and novel lineage-specific domains, emphasising the involvement of gene birth in lineage evolution on a broad scale.

## Changes in the transcription factor repertoire and in membrane processes accompany bilaterian evolution

### Nuclear factors include key developmental regulators

To reveal the putative function of the identified bilaterian-specific genes, we determined the subcellular location of their human orthologues according to the information at www.uniprot.org (*Figure 3*). Almost two-thirds of the 157 genes belonged to either of two cellular compartments, the nucleus or the plasma membrane. The majority of nuclear proteins (40/57 orthogroups) had transcription factor activity, with various domains for DNA binding (*Figure 3B*). Although $C_2H_2$ poly-ZF proteins are particularly enriched (*Figure 2—figure supplement 1*, *Supplementary file 1*–Supplementary Table 8), we also found several transcription factors with homeobox and basic helix-loop-helix (bHLH) domains (*Figure 3B*; *Figure 2*). The latter factors are important for regulatory processes during embryogenesis such as neurogenesis, myogenesis, and positional specification along the body axis (*Supplementary file 1*–Supplementary Table 10). For example, we found the bHLH domain-containing transcription factor MyoD, the master regulator for muscle cell specification in vertebrates, *D. melanogaster*, and *C. elegans* (*Tapscott et al., 1988*; *Michelson et al., 1990*; *Chen et al., 1994*), consistent with the bilaterian origin of mesoderm (*Supplementary file 1*–Supplementary Table 10, *Supplementary file 4*). Likewise, at least three conserved regulators of nervous system development and neurotransmission, the Neuronal PAS domain-containing protein 4, the Prospero homeobox protein 2, and the Achaete-scute homologue 2 (*Stergiopoulos et al., 2014*; *Sun and Lin, 2016*), emerged in the ancestor of bilaterians (*Supplementary file 1*–Supplementary Table 10, *Supplementary file 4*). Finally, two orthogroups with homeobox domain proteins, OG_8634 and OG_4203, contained the central Hox genes Antennapedia and Ultrabithorax (*Balavoine et al., 2002*; *Chourrout et al., 2006*). Central Hox genes are absent from non-bilaterian Metazoa despite the existence of anterior and posterior homologues (*Ryan et al., 2007*). Our screen thus correctly identified central Hox genes as a bilaterian novelty even though homeodomain-containing proteins are difficult to assign (*Thomas-Chollier et al., 2010*; *Hueber et al., 2013*).

### Membrane factors include neural transducers and novel proteins

A heterogeneous set of proteins was mapped to the membrane compartment (*Figure 3D*). While most of the domains found in 49 orthogroups of this category occurred once or twice, several domains were seen more often, in particular the seven transmembrane receptor domain (7tm; 13×), the leucine-rich repeat (LRR; 5×), the Bestrophin chloride channel (Bestrophin; 3×), and the hormone receptor domain (HRM; 3×). The 7tm domain is characteristic of G-protein-coupled receptors, which will be discussed further below. The LRR domain is a protein binding motif (*Kobe and Kajava, 2001*) and present in several factors connected to the plasma membrane (*Figure 3D*) such as LINGO1, SLIT2, or SEMA6C. These LRR domain-containing molecules are crucial for organising neural connectivity and are employed for axon guidance, myelination, and synapse formation (*de Wit et al., 2011*). Although LRR domain-containing molecules exist in non-bilaterians (*Ocampo et al., 2015*), it is currently unknown whether they fulfil, in these organisms, a role in nervous system development as observed in flies and vertebrates. Further, several bilaterian-specific orthogroups contained ion channel proteins. For both nervous system function and embryonic development (*Moody et al., 1991*; *Pai et al., 2017*), ion channels play important roles as they provide the basis of currents and action potentials across the plasma membrane and are involved in morphogenetic movements and cell shape changes during development (*Moody et al., 1991*). However, most ion channel proteins

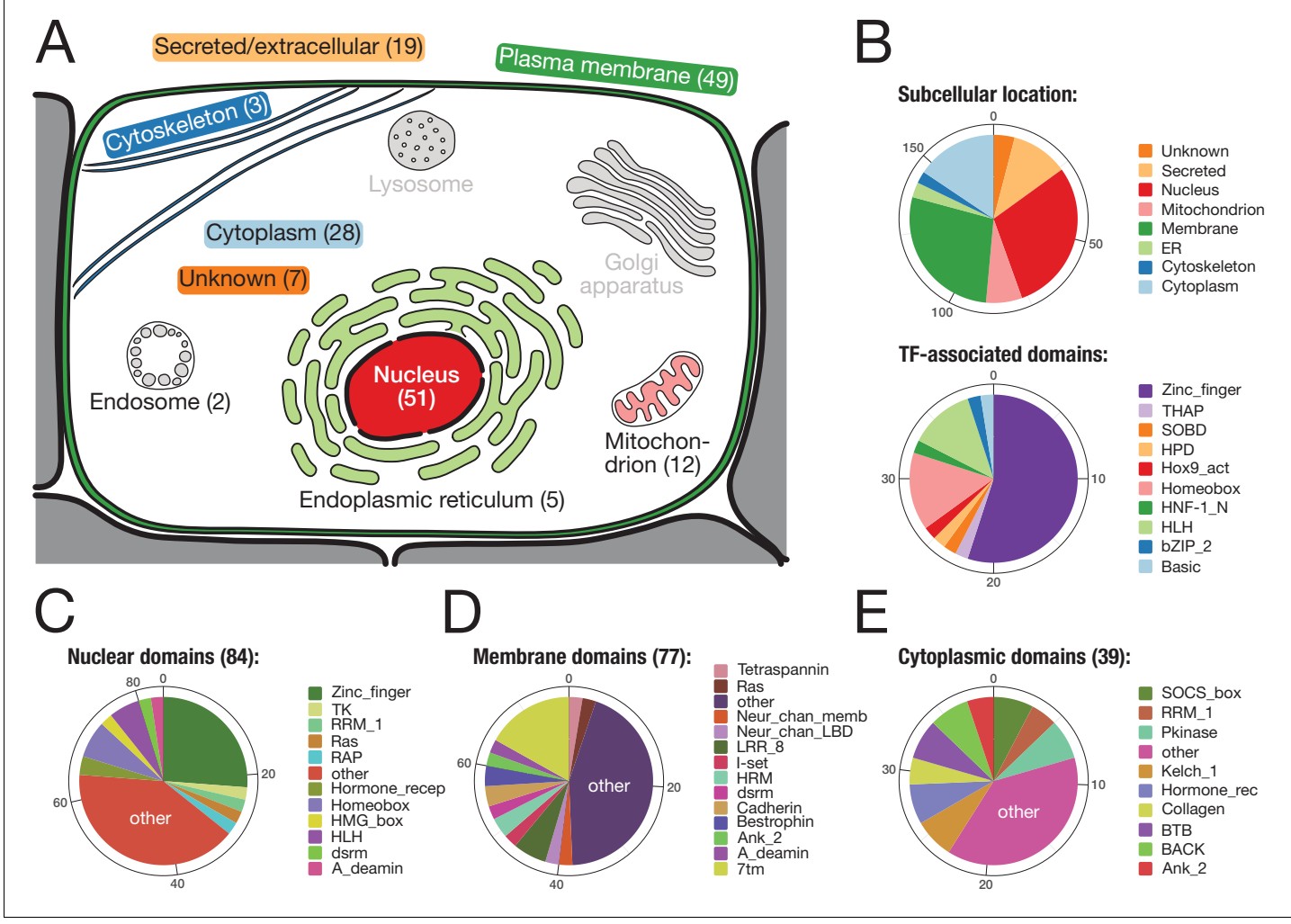

**Figure 3.** Subcellular location and molecular function of 157 bilaterian-specific genes. (**A**) Graphic representation of a eukaryotic cell with its typical organelles. Numbers in parentheses denote the number of bilaterian-specific orthogroups associated predominantly with a given cellular structure. Graphic drawn after the subcellular location section at uniprot.org. (**B**) Upper chart: Subcellular location of 157 bilaterian-specific genes. Location data is based on the corresponding human orthologues and colour-matched with the graphics in A. Lower chart: Number and name of transcription factor-associated domains present in the set of 157 bilaterian-specific genes. The 40 orthogroups are a subset of 51 orthogroups associated with the nuclear compartment. In most cases, domains names follow Pfam standards (http://pfam.xfam.org/). (**C**) Distribution of 84 domains found in 51 orthogroups associated with the nucleus. (**D**) Distribution of 77 domains found in 49 orthogroups associated with the plasma membrane. (**E**) Distribution of 39 domains found in 28 orthogroups associated with the cytoplasm. "Other" represents domains found only once in the respective category.

seem to predate the origin of metazoans (*Jegla et al., 2009*), and therefore it is unclear how the identified channel proteins affected bilaterian evolution.

Three orthogroups contained transmembrane proteins for which currently no functional description is available, although expression data for two of these exist: OG_13067 (TM169_HUMAN), OG_26661 (TM74B_HUMAN), and OG_28197 (TM160_HUMAN). Genome-wide studies revealed that CG4596, the *Drosophila* orthologue of TM169_HUMAN, is expressed in the ventral nerve cord, ventral midline, and in the brain during embryogenesis (*Tomancak et al., 2002*), similar to a central nervous system-based expression of the mouse orthologue (*Supplementary file 1*–Supplementary Table 11; *Petryszak et al., 2016*). Mouse expression data for the transmembrane protein TM160_HUMAN largely overlap with TM169_HUMAN (*Supplementary file 1*–Supplementary Table 11), but corresponding data from *Drosophila* are not available, as TM160 is absent from ecdysozoans (*Figure 2—figure supplement 2*, *Supplementary file 1*–Supplementary Table 12). Multiple sequence alignments and HMM-HMM searches demonstrate further that these two transmembrane

proteins are well conserved across bilaterians (*Figure 2—figure supplement 2*) and possess a unique sequence profile without similarity to other orthogroups within the opisthokont search space (*Supplementary file 1*–Supplementary Table 12). Together, these observations establish that so far uncharacterised proteins with predicted transmembrane domains and distinct structures might have a function in the nervous system since the Cambrian.

## Lineage-specific genes are ubiquitous and contain lineage-specific protein domains

The dataset for this study was designed to capture genes with bilaterian-specific distribution. To explore whether it allows the identification of genes specific for other evolutionary nodes, we determined the number of lineage-specific orthogroups for five successive nodes in two lineages: in the protostome lineage leading to Diptera and in the deuterostome lineage leading to Mammalia. We counted for every node lineage-specific orthogroups as a function of increasing species coverage. Extending coverage reduced the number of lineage-specific orthogroups, as expected (*Figure 4*). However, tens to hundreds of lineage-specific orthogroups were still obtained at each individual node under the strict condition of 50% coverage (i.e. at least 50% of the species that belong to the respective node need to be present in orthogroups; *Figure 4*). HMM-HMM searches and domain scans further suggested that lineage-specific orthogroups for the 10 nodes contain novel domains unique to the respective lineage (for examples, see *Figure 4—figure supplement 1* and *Supplementary file 1*–Supplementary Table 13), as it is the case for bilaterian-specific proteins (*Figure 2—figure supplement 2*, *Figure 2—figure supplement 3*, *Figure 2—figure supplement 4*). These findings suggest that the origin of genes and novel protein domains is a robust component of evolution at every examined node and that the faithful identification of these genes is a critical aspect in reconstructing evolutionary history, as exemplified by the recent detection of lineage-specific genes in mammals, mollusks, cnidarians, or arthropods (*Milde et al., 2009*; *Aguilera et al., 2017*; *Dunwell et al., 2017*; *Thomas et al., 2020*).

## The Nodal pathway is a bilaterian-specific addition to the TGF-β superfamily and linked to left-right determination and mesoderm formation

Three orthogonal axes—the anterior-posterior, the dorsal-ventral, and the left-right axis—determine body layout in bilaterian animals. One of the signalling systems active in these processes is the Nodal pathway. It belongs to the transforming growth factor β (TGF-β) pathway and is essential for the specification of left-right asymmetry and the induction of mesoderm and endoderm in vertebrates (*Shen, 2007*). The TGF-β ligands Nodal and Lefty, the co-receptor EGF-CFC, and the transcription factor FoxH1 are components specific to the Nodal pathway (*Figure 5—figure supplement 1*). In addition, the T-box transcription factor TBR-2/Eomes (T-box brain protein 2/Eomesodermin) is a target of Nodal signalling and critical for mesoderm formation and neural development (*Ryan et al., 1996*; *Arnold et al., 2008*).

Distinct phylogenetic distributions have been reported for the Nodal-signalling components. The presence and functional conservation of Nodal itself is well established across deuterostomes (*Duboc et al., 2004*; *Hudson and Yasuo, 2005*; *Shen, 2007*; *Röttinger et al., 2015*) and lophotrochozoans (*Grande et al., 2014*; *Kenny et al., 2014*). In contrast, searches for Lefty orthologues were so far positive only in deuterostomes (*Chen and Schier, 2002*; *Mita and Fujiwara, 2007*; *Duboc et al., 2008*; *Li et al., 2017*), but not in Lophotrochozoa (*Grande et al., 2014*). Similarly, the Nodal coreceptor EGF-CFC has been identified only in deuterostomes (*Yan et al., 1999*; *Ravisankar et al., 2011*), and FoxH1 orthologues have been characterised in vertebrates and cephalochordates only (*Weisberg et al., 1998*; *Zhou et al., 1998*; *Yu et al., 2008*; *Figure 5A*). Nodal-signalling components have not been identified in the protostome model organisms *D. melanogaster* and *C. elegans*. Likewise, the T-box factor *eomesodermin* is absent from these animals but has been described in lophotrochozoans, deuterostomes, and sponges (*Maruyama, 2000*; *Tagawa et al., 2000*; *Arenas-Mena, 2008*; *Arnold et al., 2008*; *Sebé-Pedrós et al., 2013*). These findings imply a successive gain of Nodal signalling components along the lineage from the metazoan to the vertebrate ancestor (*Figure 5A*).

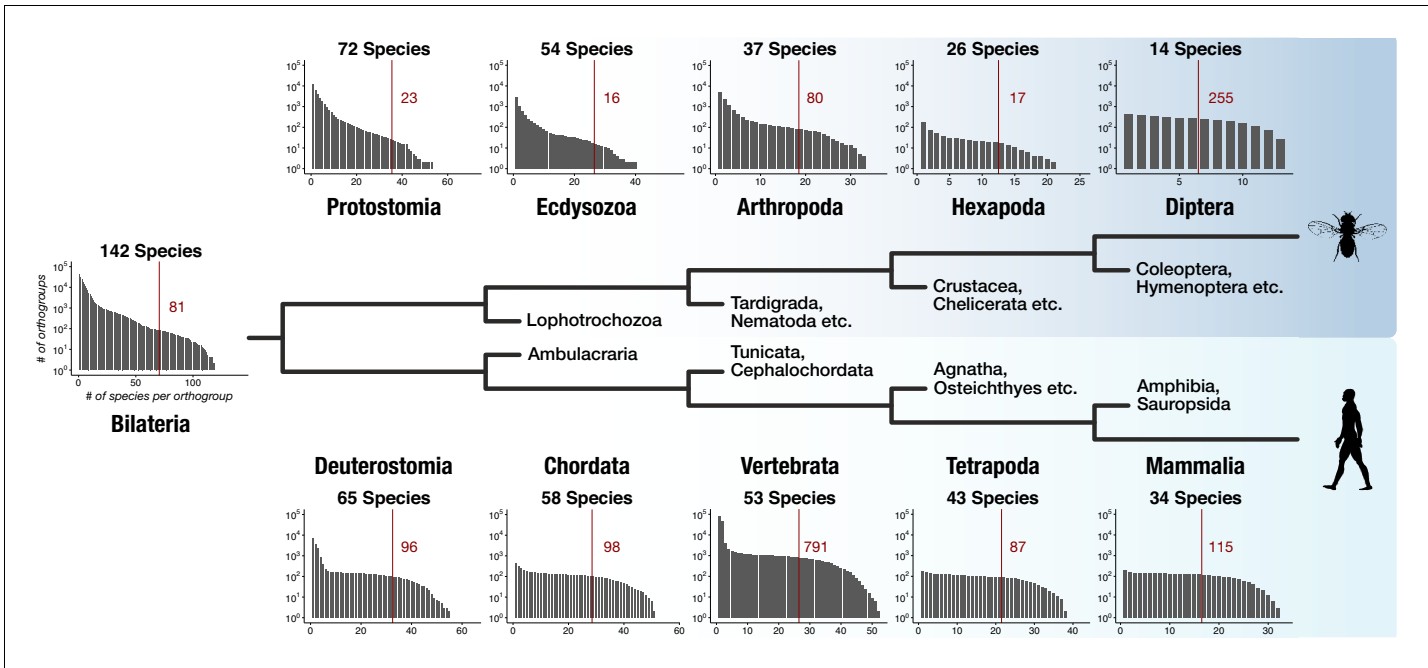

**Figure 4.** Distinct lineage-specific genes at subsequent nodes of insect and vertebrate evolution. Starting from Bilateria (left), a protostome lineage leading to dipterans (upper) and a deuterostome lineage leading to mammals (lower) are shown as schematic phylogenetic tree. Sister clades to the selected taxa are denoted on short branches in the center. Each barplot displays the number of lineage-specific orthogroups (y axis) as a function of orthogroup size (x axis) for the selected taxonomic group (Protostomia, Ecdysozoa, Arthropoda etc.). The total species count (within BigWenDB) for each of the eleven taxonomic groups is indicated on top of the corresponding barplots (# Species). The count of lineage-specific genes decreases with growing orthogroup size. A red line denotes the number of orthogroups in which at least 50% of the species of a selected lineage are present. The corresponding number of lineage-specific orthogroups is highlighted in red next to the line.

The online version of this article includes the following figure supplement(s) for figure 4:

**Figure supplement 1.** Exemplary multiple sequence alignments of three arthropod-specific orthogroups without known domains.

In line with previous findings (*Hudson and Yasuo, 2005*; *Shen, 2007*; *Grande et al., 2014*; *Kenny et al., 2014*), our analysis revealed that the TGF-β ligand Nodal belongs to a robust bilaterian-specific orthogroup (OG_12210; *Figure 5—figure supplement 2*, *Supplementary file 1*–Supplementary Table 14). However, orthogroups of the other Nodal pathway members (Lefty, EGF-CFC, FoxH1, and Eomes) were also bilaterian-specific, and HMM-HMM-based searches against all orthogroups (*Supplementary file 1*–Supplementary Table 14) as well as phylogenetic analyses supported this result (*Figure 5—figure supplement 2*, *Figure 5—figure supplement 3*).

Our clustering results suggested further that the T-box transcription factor Eomes is in fact restricted to bilaterians, contradicting a study that identified Eomes candidates in two poriferan species (*Sebé-Pedrós et al., 2013*). In BLAST searches, the two poriferan sequences displayed highest similarity to the canonical T-box transcription factors TBX3/4, but not to the T-box containing protein Eomes (*Supplementary file 1*–Supplementary Table 15). Likewise, phylogenetic analyses failed to confidently assign the poriferan sequences to the Eomes clade (*Figure 5—figure supplement 4*), and HMM-HMM searches could not detect Eomes-related orthogroups with proteins from sponges or other non-bilaterian animals (*Supplementary file 1*–Supplementary Table 14). These results consistently argue for a bilaterian origin of the factor, matching the distribution of the other Nodal pathway members (*Figure 5B*). While our phylogenetic analyses supported orthology clustering results and the monophyly of the Eomes clade, they unexpectedly argued for a metazoan origin of the gene (*Figure 5—figure supplement 4*). This interpretation would imply independent loss events in the ancestors of three phyla (Cnidaria, Placozoa, and Ctenophora) and in two sponge lineages (see *Figure 5A* and discussion), while a posited bilaterian-specific origin would be more parsimonious. To finally resolve this issue, more detailed analyses are needed.

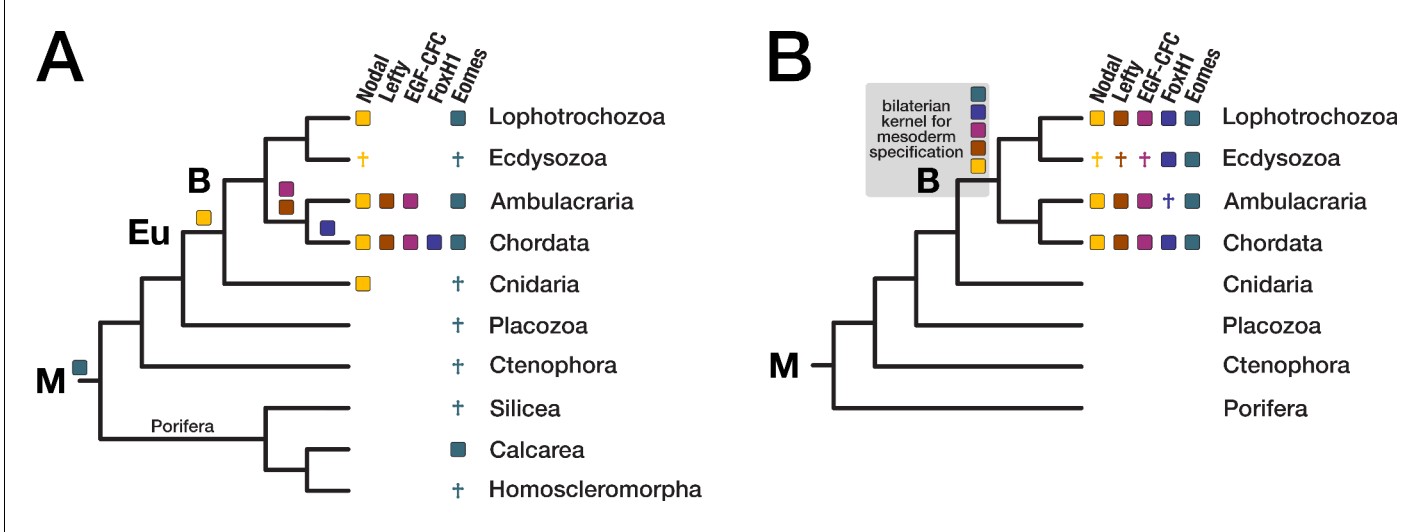

**Figure 5.** Evolution of the Nodal signaling pathway. Two consensus phylogenetic trees showing the relationship of major metazoan lineages. The five factors of the Nodal signalling pathway (Nodal, Lefty, EGF-CFC, FoxH1, and Eomes) are displayed as coloured boxes. Their phylogenetic distribution and inferred evolutionary origin are mapped onto the tree. Gene births are indicated as coloured boxes above the respective branch. Inferred losses are represented by crosses. Bold labels to the left of a branch indicate branch ancestors: B = Bilateria, Eu = Eumetazoa, M = Metazoa. (**A**) Previous results regarding the evolution of Nodal pathway genes, as known from the literature. (**B**) Revised evolutionary history of the Nodal pathway genes according to our results. Note that none of the five factors has been found in arthropods and nematodes. The ecdysozoan boxes for Eomes and FoxH1 are derived from the presence of the genes in a single priapulid species. Grey shading: Hypothetical emergence of a putative kernel for mesoderm specification and neural patterning.

The online version of this article includes the following figure supplement(s) for figure 5:

**Figure supplement 1.** Schematic outline of the Nodal signaling pathway in vertebrates.

**Figure supplement 2.** Bilaterian-specific distribution of the Nodal pathway components Nodal and Lefty.

**Figure supplement 3.** Bilaterian-specific distribution of the Nodal pathway component FoxH1.

**Figure supplement 4.** Bilaterian-specific distribution of the Nodal pathway component Eomesodermin.

Recently, a Nodal-related gene has been identified in the cnidarian *Hydra magnipapillata* and found to be essential for specifying axial asymmetry along the polyp's main body axis (*Watanabe et al., 2014*). In our dataset, *H. magnipapillata* Nodal-related belongs to a different orthogroup (OG_9136), together with sequences from nine other cnidarians and many deuterostomes. This orthogroup contains, among others, vertebrate GDF-6/7, but no Nodal orthologues. Furthermore, we did not obtain an HMM-HMM reciprocal best hit relationship with the Nodal orthogroup using as query either the entire orthogroup OG_9136 or a subset of cnidarian sequences (*Supplementary file 1*–Supplementary Table 16), suggesting that Nodal indeed emerged in the bilaterian ancestor as a new member among pre-existing Nodal-related genes.

Taken together, orthology clustering, HMM-HMM comparison, and phylogenetic evidence establish that all four Nodal-specific pathway components and Eomes are present only in bilaterians (*Figure 5B*). It is thus possible that these factors co-evolved as extension of the more ancient TGF-β signalling pathway (*Huminiecki et al., 2009*; *Hinck et al., 2016*) and acquired the potential for mesoderm formation and left-right axis determination, two characteristic bilaterian traits. Due to the conservation of this hypothetical gene regulatory network (GRN) since the Cambrian, it could represent an ancient kernel for mesoderm specification and neural patterning. The identification of only a subset of the five factors in non-chordate species (*Figure 5B*) indicates that Nodal signalling experienced substantial evolutionary turnover, but it does not exclude initial assembly of the pathway in the bilaterian ancestor and subsequent lineage-specific changes.

One consequence of these considerations is that large parts of the Nodal GRN must have been lost early in ecdysozoan evolution, implying the evolution of alternative upstream signalling pathway inputs for axial specification in this group. Secondly, genes that originated in the bilaterian ancestor may have been lost in a particular daughter lineage. The widespread loss of genes across metazoans (*Richter et al., 2018*; *Sharma et al., 2018*) and the loss of Nodal pathway members (this study)

shows that such scenarios are conceivable and might impact the exhaustive description of lineage-specific genes, that is, the reconstruction of the "true" evolutionary history of a taxon.

## G-protein-coupled receptors and the control of physiological state through circulatory flow

Among the identified bilaterian-specific genes is a set of eight G-protein-coupled receptors (GPCRs), members of a large family of seven-transmembrane domain receptors. While GPCRs are ancient and were already present in the ancestor of bilaterians and fungi (*Krishnan et al., 2012*), our results indicate that new members of the GPCR family appeared at the bilaterian base. Specifically, robust clustering results and HMM-HMM comparisons place the origin of monoamine neurotransmitter receptors for serotonin, adrenaline, and dopamine to the bilaterian root (*Supplementary file 1*–Supplementary Table 17, *Supplementary file 1*–Supplementary Table 18), in line with a recent publication that dated back the evolutionary history of adrenergic signalling to the bilaterian ancestor (*Bauknecht and Jékely, 2017*). Histochemical, biochemical, and functional data are in conflict with this finding and argue for the presence of serotonin, dopamine, and other small molecule neurotransmitters in cnidarians, the bilaterian sister group (*Carlberg and Anctil, 1993*; *Kass-Simon and Pierobon, 2007*; *Mayorova and Kosevich, 2013*). However, receptors for these molecules could not be identified unambiguously in cnidarians (*Anctil, 2009*; *Bosch et al., 2017*), maintaining the possibility that they indeed constitute bilaterian innovations.

There is evidence across several bilaterian phyla (arthropods, nematodes, mollusks, platyhelminthes, vertebrates) that adrenaline, dopamine, and serotonin signalling regulates many important processes such as behaviour, feeding, learning, locomotion, memory, reproduction, reward, or sleep (*Ségalat et al., 1995*; *Berridge, 2004*; *Suo et al., 2004*; *Berger et al., 2009*; *Vidal-Gadea et al., 2011*; *Burke et al., 2012*; *El-Shehabi et al., 2012*; *Ueno et al., 2012*). In addition to these "post-embryonic" functions, serotonin is recognised as an important regulator of embryonic development and neuronal circuitry in vertebrates and invertebrates (*Brown and Shaver, 1989*; *Buznikov et al., 2001*; *Daubert and Condron, 2010*). The proposed origin of monoamine neurotransmitter receptors in the bilaterian ancestor (*Supplementary file 1*–Supplementary Table 17, *Supplementary file 1*–Supplementary Table 18) and the related functions of monoamine neurotransmitter signalling across phyla suggest that diverse functions of monoamine neurotransmitter signalling already existed in the bilaterian ancestor and could have played a role in the evolution of complex development, brain function, and behaviour. Preliminary evidence indicates that cnidarians, as the bilaterian sister group, do not respond to rewarding or punishing stimuli as do bilaterians (*Barron et al., 2010*). A link between this behavioural difference and the evolution of monoamine neurotransmitter receptors would comply with the previous notion that the evolution of dopamine-based brain reward systems in bilaterians started from dopamine's ancient role as a signalling molecule for motor circuits (*Barron et al., 2010*).

In addition to monoamine neurotransmitter receptors, we detected several peptide hormone receptors in the set of bilaterian-specific GPCRs and could support their bilaterian origin using HMM-HMM searches: the receptors for secretin, corticotropin-releasing factor, neuromedin-U, calcitonin, and somatostatin (*Supplementary file 4*, *Supplementary file 1*–Supplementary Table 17, *Supplementary file 1*–Supplementary Table 18). In vertebrates, these GPCRs and their hormone ligands are part of the endocrine system and regulate basal physiological activities such as feeding, energy homoeostasis, or stress (*Budhiraja and Chugh, 2009*; *Afroze et al., 2013*). homologues of the five receptors and their ligands have also been described in *C. elegans* and *D. melanogaster* (*Johnson et al., 2005*; *Cardoso et al., 2006*; *Melcher et al., 2006*; *Lindemans et al., 2009*; *Cardoso et al., 2014*; *Kunst et al., 2014*; *Ketchesin et al., 2017*), and the putative bilaterian ancestry of some of these signalling systems has been recognised by others, in agreement with our results (*Johnson et al., 2005*; *Lindemans et al., 2009*; *Mirabeau and Joly, 2013*). In contrast to vertebrates or insects, cnidarians and other non-bilaterian Metazoa do not contain specialised endocrine organs and circulatory systems. Thus, our finding of highly conserved peptide hormone receptors supports the view that major physiological regulators evolved in parallel with the emergence of circulatory systems. Moreover, recent evidence indicates that these hormone receptors also act during development and participate in neuronal migration and nervous system formation (*Afroze et al., 2013*; *Liguz-Lecznar et al., 2016*; *Galas et al., 2017*), suggesting an ancient link between the

generation of complex nervous systems and the ability to control body functions through circulatory fluid.

## Changes in axon guidance accompany bilaterian evolution

Axon guidance, the guided outgrowth of axons and dendrites, is essential for the development of neuronal connections and mediated by two major pathways, the Netrin-DCC and the Slit-Robo (Round-About) pathway (*Lowery and Van Vactor, 2009*; *Evans, 2016*). To reveal whether changes in these processes accompanied the evolution of bilaterians, we studied the respective orthogroups. Except one, all human Netrin paralogues were assigned to a single orthogroup. Its composition and the composition of its HMM-HMM best hit orthogroups support the emergence of Netrins in the ancestor of eumetazoans or earlier (*Supplementary file 1*–Supplementary Table 19), in line with a description of Netrins in the sea anemone *N. vectensis* (*Putnam et al., 2007*). We found a corresponding (eu)metazoan origin for the Netrin receptor DCC (*Supplementary file 1*–Supplementary Table 19). These results indicate that cnidarians, but not ctenophores, might regulate axon outgrowth at least in part by Netrin-DCC based interactions, consistent with an independent origin of the nervous system in ctenophores (*Moroz et al., 2014*).

Although orthogroup composition of Slit and its receptor Robo suggested a bilaterian origin of this system, reciprocal HMM-HMM searches indicated the existence of cnidarian Robo orthologues that were assigned to a separate orthogroup, OG_51853 (*Supplementary file 1*–Supplementary Table 19). Like their bilaterian counterparts, the cnidarian Robo candidates had highly disordered cytoplasmic domains, as revealed by structure predictions of the extracellular and intracellular part of representative sequences (*Figure 6*). On the other hand, sequence comparisons revealed that the conserved cytoplasmic motif CC1, which is required for binding the Ena/VASP protein Enabled and for transducing signals to the actin cytoskeleton (*Bashaw et al., 2000*), is altered in cnidarian Robos (*Figure 6—figure supplement 1*), and that cnidarian Robos displayed several insertions and deletions in the cytoplasmic part when compared with bilaterian Robos (*Figure 6—figure supplement 2*). It is therefore an open question whether the structural differences in cnidarian Robo-like proteins involve interactions with different downstream partners and whether cnidarian Robos regulate axon growth. Known downstream effectors of Robo signalling, such as Enabled and Son of sevenless, originated early in metazoan evolution (*Supplementary file 1*–Supplementary Table 20) and could provide in principle the functionality for Robo-based axon guidance, although mediated by a different ligand.

In both *Drosophila melanogaster* and vertebrates, midline glia cells secrete the Slit protein to prevent Robo expressing axons from crossing the body midline (*Rothberg et al., 1990*; *Brose et al., 1999*; *Kidd et al., 1999*), indicating that a key component in the establishment of bilaterally symmetric nervous systems is shared between protostomes and deuterostomes. However, in our dataset, a single placozoan sequence was assigned to Slit's otherwise bilaterian-specific orthogroup, shifting its origin back in time. BLAST searches at NCBI verified a reciprocal best hit relationship of the putative placozoan Slit to known Slit proteins, in agreement with our clustering results (*Supplementary file 1*–Supplementary Table 15). Likewise, placement of the placozoan sequence in phylogenetic analyses is compatible with its orthology to the Slit protein (*Figure 6—figure supplement 3*). Unexpectedly, HMM-HMM comparisons could not reveal the existence of Slit in other non-bilaterian species such as cnidarians or ctenophores (*Supplementary file 1*–Supplementary Table 21). From these results, we conclude that Slit and Robo probably originated in the common ancestor of placozoans, cnidarians, and bilaterians. However, the Slit-Robo-based mechanism for midline repulsion during nervous system development appears to be restricted to bilaterians, as placozoans lack a nervous system and cnidarians lack the Slit ligand.

## Neurotrophin receptor signalling is a bilaterian innovation

Neurotrophin signalling plays a fundamental role in nervous system generation by regulating many aspects of neuronal development and function, such as neuronal survival, synapse formation, or axon guidance (*Huang and Reichardt, 2001*; *Lu et al., 2005*). Vertebrates possess four related neurotrophin ligands and three corresponding transmembrane receptors of the Trk family that each originated from a single ancestral gene in chordates (*Benito-Gutiérrez et al., 2005*; *Hallböök et al., 2006*). Once considered a vertebrate innovation, neurotrophins and their receptors have now been

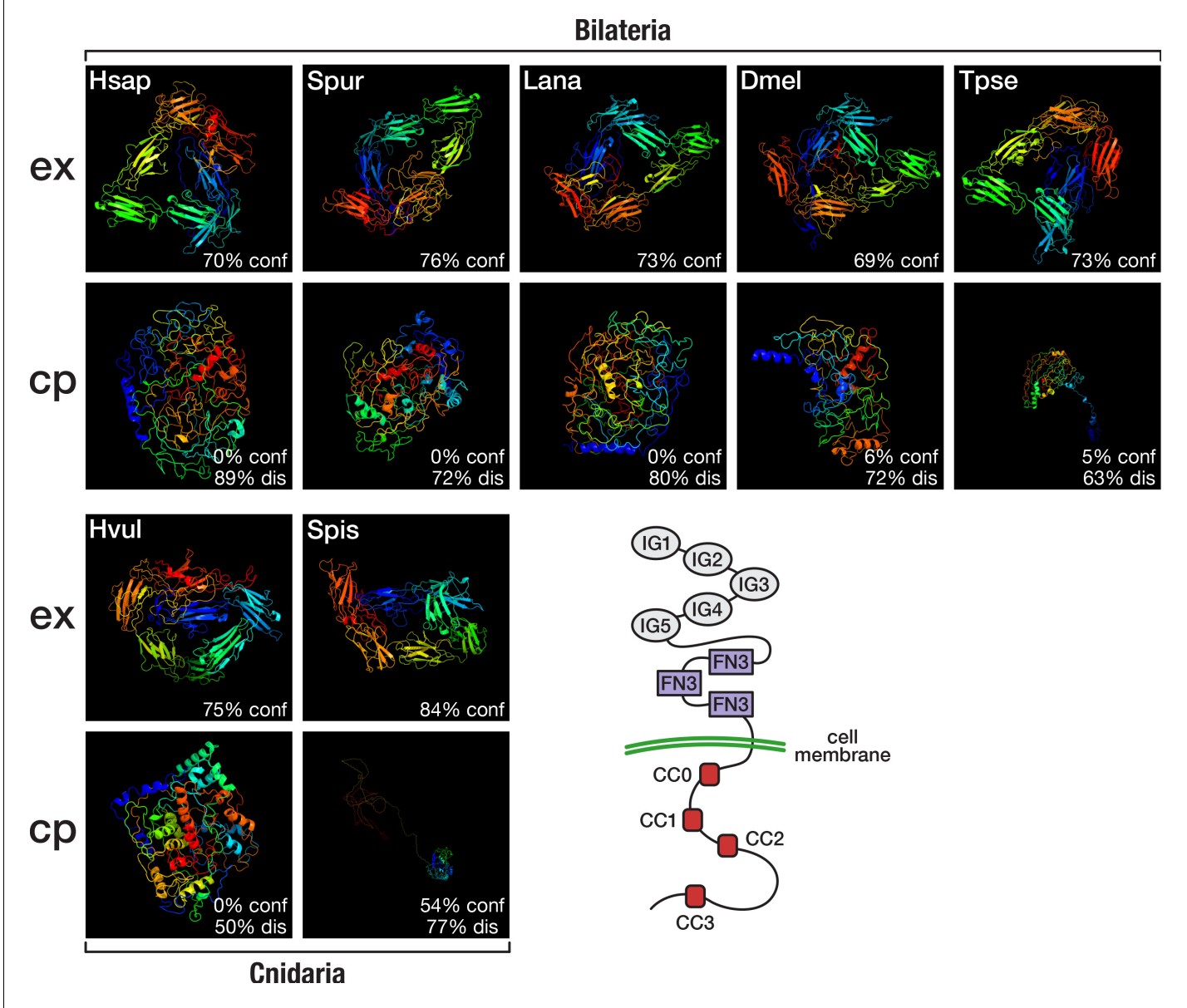

**Figure 6.** Structural predictions of cnidarian and bilaterian Robo proteins. Top (ex): Predicted structure of the extracellular domain plus transmembrane region of seven selected Robo proteins. Bottom (cp): Predicted structure of the transmembrane region plus cytoplasmic part of seven selected Robo proteins. Robo1 orthologues of two deuterostomes (Hsap = *Homo sapiens*; Spur = *Strongylocentrotus purpuratus*), one lophotrochozoan (Lana = *Lingula anatina*), two ecdysozoans (Dmel = *Drosophila melanogaster*; Tpse = *Trichinella pseudospiralis*), and two cnidarians (Hvul = *Hydra vulgaris*; Spis = *Stylophora pistillata*) were analysed. "% conf" indicates the percentage of residues modelled at >90% confidence. "% dis" indicates the predicted percentage of disordered regions. Bottom right: Schematic outline of the Robo domain structure with five immunoglobulin domains (IG1–IG5) and three fibronectin type III domains (FN3) in the extracellular part and four conserved cytoplasmic motifs (CC0–CC3) in the intracellular part. Like their bilaterian counterparts, cnidarian Robo candidates display a disorganised protein structure in the cytoplasmic part despite differences in structural features (*Figure 6—figure supplement 1*, *Figure 6—figure supplement 2*). The extracellular part (top row), on the other hand, is similarly organised across metazoans.

The online version of this article includes the following figure supplement(s) for figure 6:

**Figure supplement 1.** Change of the conserved cytoplasmic motif CC1 in cnidarian Robo-like proteins.

**Figure supplement 2.** Cnidarian Robo-like proteins display structural alterations.

**Figure supplement 3.** Phylogenetic analysis of a putative *Trichoplax adhaerens* Slit protein.

found in diverse invertebrates (*Wilson, 2009*; *Kassabov et al., 2013*; *Lauri et al., 2016*). In particular, studies in the mollusk *Aplysia californica* suggest that neurotrophin signalling and neurotrophin-mediated synaptic plasticity are conserved in protostomes and deuterostomes (*Kassabov et al., 2013*).

To elucidate the evolutionary origin of neurotrophin signalling, we analysed the orthogroups containing neurotrophins and their receptors. The four vertebrate neurotrophin ligands clustered into two bilaterian-specific orthogroups (OG_14798 and OG_21801) that are each other's reciprocal best hit. We could not detect orthogroups similar to neurotrophins in non-bilaterian metazoans or additional, so far unidentified neurotrophins in bilaterians (*Supplementary file 1*–Supplementary Table 22), supporting the emergence of a single neurotrophin gene in the ancestor of bilaterians and its subsequent diversification in vertebrates. When we analysed the evolutionary origin of other neurotrophic factors, we recognised that they also arose in the ancestor of bilaterians or even later (*Figure 7*; *Supplementary file 1*–Supplementary Table 22, *Supplementary file 1*–Supplementary Table 23). The evolutionary age of these additional neurotrophic factors is thus consistent with a bilaterian origin of neurotrophic ligands per se. The same evolutionary scenario is supported by detailed analysis of the Trk receptor family. Although our initial dataset conflated Trk and Wnt pathway receptors due to a shared receptor tyrosine kinase domain, adjustment of the MCL inflation parameter successfully rendered a Trk-only orthogroup, whose taxonomic composition is restricted to bilaterians (*Figure 7—figure supplement 1*; *Supplementary file 1*–Supplementary Table 24).

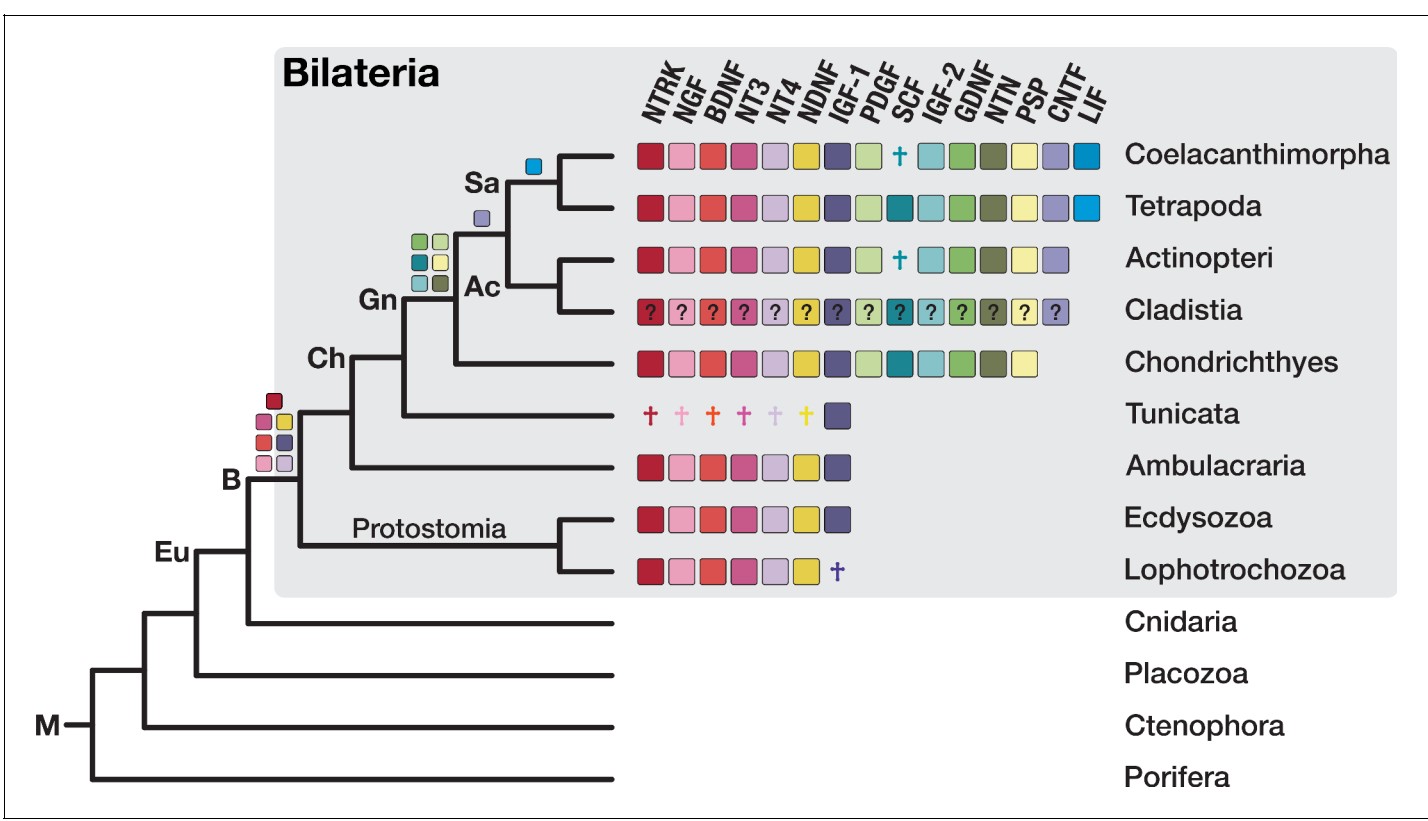

**Figure 7.** The bilaterian-wide distribution of neurotrophic factors. The NTRK receptor and 14 major neurotrophic factors are displayed as coloured boxes. Their phylogenetic distribution and inferred evolutionary origin are mapped onto the tree (see *Supplementary file 1*–Supplementary Table 22 and *Supplementary file 1*–Supplementary Table 23). Gene births are indicated as coloured boxes above the respective branch of the tree (left). Inferred losses are shown as coloured crosses in the matrix. Bold labels to the left of a branch indicate branch ancestors: Ac = Actinopterygii, B = Bilateria, Ch = Chordata, Eu = Eumetazoa, Gn = Gnathostomata, M = Metazoa, Sa = Sarcopterygii. The neurotrophic factors of Cladistia, the sister group of Actinopteri, are inferred and distinguished by a question mark as the dataset lacks species from this lineage.
The online version of this article includes the following figure supplement(s) for figure 7:

**Figure supplement 1.** The NTRK neurotrophin receptor is restricted to bilaterians.

These results indicate that neurotrophins and their receptors are present across bilaterians and might fulfill conserved functions in neuronal development in these animals. If long-term potentiation and memory formation is regulated by serotonin and its receptors across bilaterians (see, for example, *Teixeira et al., 2018*), a link between serotonin action and neurotrophin signalling may have emerged in the bilaterian ancestor that contributed to nervous system evolution and the learning-dependent synaptic plasticity characteristic for this group.

## Bilaterian-specific factors and the evolution of excretory systems

Protostomes and deuterostomes comprise the taxon Nephrozoa, animals with a dedicated excretory system (sensu *Jondelius et al., 2002*). Together with their sister group Xenacoelomorpha, Nephrozoa form the taxon Bilateria (*Cannon et al., 2016*). When we started with our study, sequences from Xenacoelomorpha were not available, and therefore our bilaterian-specific gene set is in fact specific for nephrozoans and might contain factors related to kidney and/or nephron development. Indeed, we identified in the 157 bilaterian-specific orthogroups two relevant zinc finger transcription factors. The poly-zinc finger transcription factor Evi1/MECOM was assigned to a large orthogroup with protein members from 108 of 142 bilaterian species (OG_5543). Evi1 is expressed in pronephric tissue of *Xenopus* and zebrafish embryos and involved in nephron patterning in these species (*Mead et al., 2005*; *Li et al., 2014*; *Desgrange and Cereghini, 2015*), although this might only be a part of its function (*Goyama et al., 2008*). Secondly, after BLAST searches, maximum likelihood phylogenetic analysis, and HMM-HMM searches focusing on orthogroup OG_5226, we found evidence for a bilaterian-wide distribution of odd-skipped related 1, a zinc finger transcription factor required for heart and urogenital development in vertebrates (*Wang et al., 2005*; *Dressler, 2006*; *Tena et al., 2007*; *Supplementary file 1*–Supplementary Table 15, *Supplementary file 1*–Supplementary Table 26; *Supplementary file 1*–Supplementary Figure 1). Thus, the observed expansion of the zinc finger transcription factor repertoire may also have been important for the evolution and development of excretory organs, a key nephrozoan innovation.

## Bilaterian-specific genes form a rich interaction network with interconnected subnetworks

To reveal potential interactions among the 157 bilaterian-specific proteins, we analysed the interaction network of the corresponding human orthologues using the STRING protein-protein interaction (PPI) database. The obtained PPI network contained significantly more interactions than expected by chance (PPI enrichment $p$-value: $5.93e^{-14}$), revealing that bilaterian-specific genes form a dense network in which about 50% of the factors (83 distinct factors) are connected to one another (*Figure 8A*). These interactions form several subnetworks involved in regulating key aspects of bilaterian development, such as chromatin organisation and transcriptional regulation (subnetwork A), myogenesis (subnetwork B), mesoderm formation and left-right asymmetry (the Nodal pathway, subnetwork C: see also *Figure 8B*), neurogenesis (subnetwork D), and physiology (subnetwork E). Connections between different subnetworks further suggest that crosstalk between the newly established regulatory subnetworks was an important aspect of bilaterian evolution.

Previous work found that protein network connectivity (number of interactions) increases with gene age (*Kim and Marcotte, 2008*). To analyse the degree of connectivity of our bilaterian network, we compared it to a PPI network generated from metazoan-specific proteins that is expected to show higher connectivity due to the proteins' more ancient origin. Our orthology clustering data identified 797 metazoan-specific proteins (>5× as many proteins as in the bilaterian dataset), and the combined bilaterian-metazoan PPI network comprised 2,531 interactions among 823 proteins (16% bilaterian-specific proteins, 84% metazoan-specific proteins). In fact, we obtained a slightly higher level of connectivity for the younger, bilaterian proteins (*Figure 8C*: total number of interactions per protein, median ± median absolute deviation (MAD): 5 ± 4.62 for Bilateria, 4 ± 4.16 for Metazoa; Mann-Whitney U test: $U = 39792, p = 0.0135$). Furthermore, bilaterian-specific proteins preferentially interacted with one another, with over twice as many bilaterian-bilaterian interactions as would be expected by chance ($\tilde{\chi}^2$ statistic = 24.814, $p = 0.000001$), primarily due to fewer bilaterian-metazoan interactions than would be expected. This is also evident at the level of individual proteins: bilaterian-specific proteins have significantly more bilaterian interaction partners (*Figure 8D*:

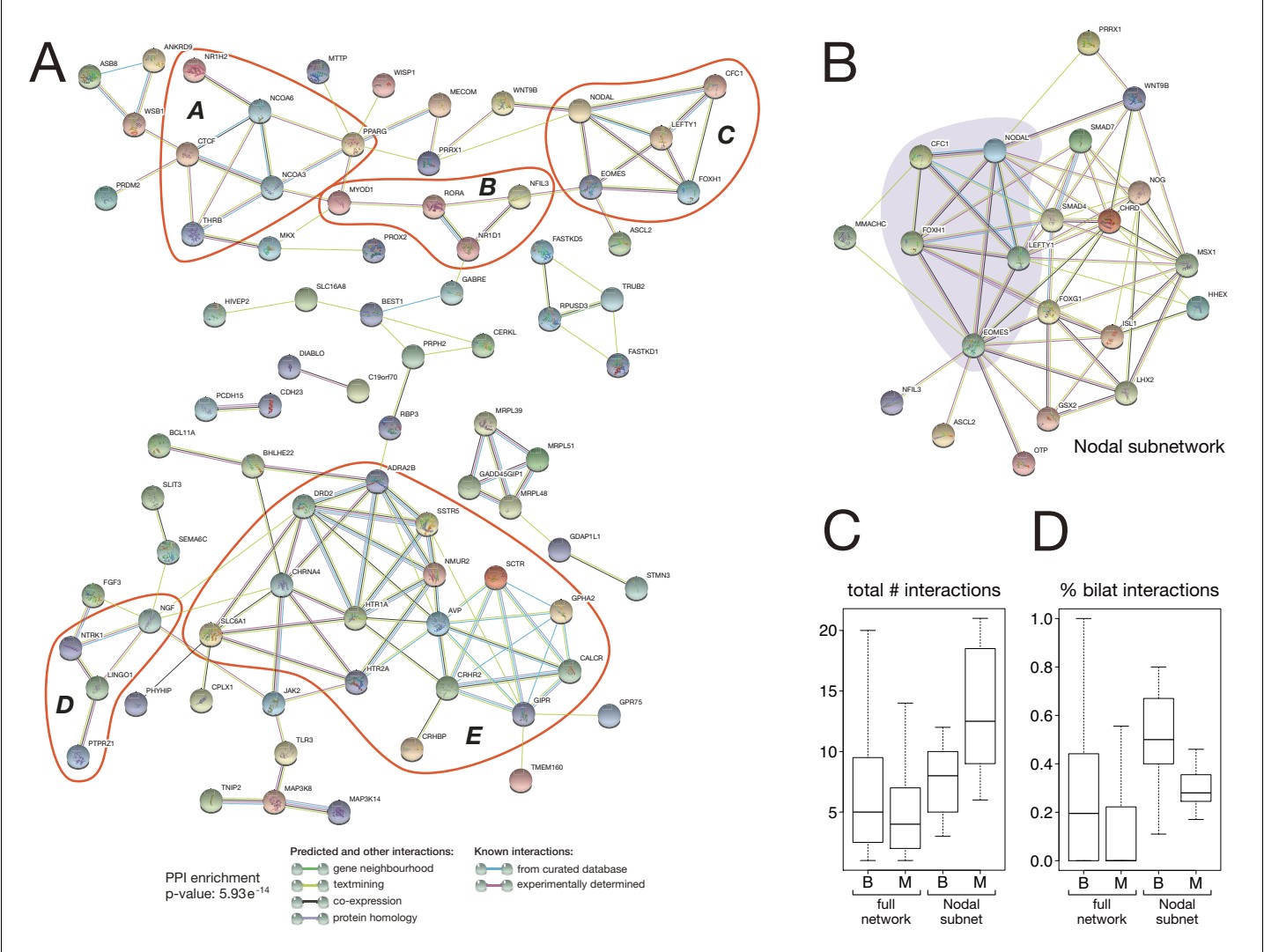

**Figure 8.** Protein-protein interaction network of bilaterian-specific proteins. (**A**) Uniprot identifiers corresponding to the human orthologues of 150 bilaterian-specific genes (seven OGs had no human orthologues) were uploaded to the STRING database, and their mutual interactions were visualised as a network. Parameters for the displayed PPI network were: minimum required interaction score = 0.4; maximum number of interactors to display in 1st and 2nd shell = 0. Thus, only known and predicted interactions between 83 distinct bilaterian-specific proteins are shown (non-interacting proteins are hidden). Evidence for displayed interactions is colour-coded (see legend). Edge length and node placement are arbitrary. Five subnetworks between bilaterian-specific genes are highlighted in red (*A-E*, see Results). (**B**) Bilaterian-specific Nodal subnetwork in the context of metazoan genes. The five members of the Nodal pathway are highlighted by shading. (**C**, **D**) Boxplots comparing bilaterian- (B) and metazoan-specific (M) proteins in the full network and Nodal subnetwork for the total number of interactions per protein (**C**), and for the relative fraction of bilaterian interactions per protein (**D**).

percent of bilaterian interactions, median ± MAD: $19.5\% \pm 23.2\%$ for Bilateria, $0.0\% \pm 16.1\%$ for Metazoa; Mann-Whitney $U = 32231$, $p = 0.00000$).

As we identify the Nodal pathway as a key bilaterian innovation (*Figure 5*, *Figure 8A*: subnetwork C), we focused on this subnetwork as a case study for further analysis of molecular interactions. Within the full bilaterian-metazoan PPI network, we indeed recovered the Nodal pathway as a bilaterian-specific subnetwork, embedded among connections to additional bilaterian and metazoan proteins (*Figure 8B*). As with the full network, for this subnetwork we found a significant number of bilaterian-specific protein interactions (*Figure 8D*; Kruskal-Wallis $\tilde{\chi}^2 = 62.855$, degrees of freedom = 3, p = 1.44e$^{-13}$). Furthermore, for this subnetwork, we found support for the hypothesis that older (metazoan) genes have higher connectivity (*Figure 8C*; *Kim and Marcotte, 2008*). Notably,

metazoan-specific proteins that participate in the Nodal subnetwork are a non-representative sub-set, showing significantly higher overall connectivity and bilaterian-specific connectivity than meta-zoan proteins in the full bilaterian-metazoan PPI network. Thus, it may be that older genes have higher connectivity if they exceed a minimum threshold of connectivity (number of interactions). For example, the Nodal subnetwork includes Smad4, a metazoan-specific protein with the highest con-nectivity (46 interactions) of any protein in our combined network. This multifunctional BMP pathway component likely exemplifies two evolutionary trends: that highly connected genes are most likely to acquire new interaction partners, and that bilaterian-specific PPI innovations build on more ancient, preexisting PPI networks by co-option.

Extrapolating these findings to interactions with additional factors of more ancient origin implies that the evolution of new genes in the bilaterian ancestor affected a large number of processes in animal biology.

## Discussion

### An R-based OrthoMCL pipeline for processing large datasets

Explaining the sudden emergence of bilaterally symmetric animals during the Cambrian is a central problem in evolutionary biology. Complicated by the uneven coverage of the metazoan tree with sequence information, a systematic approach to identify the genetic basis for the evolution of bilat-erians was missing. In this study, we present a comparative genomics approach, designed to provide maximum resolution at the bilaterian/non-bilaterian divergence and therefore uniquely suited to dis-cover bilaterian-specific genes.

Although sequence data for individual species in our study might be incomplete (*Supplementary file 1*–Supplementary Table 1, *Supplementary file 1*–Supplementary Table 2), each important taxonomic group (Deuterostomia, Ecdysozoa, Lophotrochozoa, and "non-Bilateria") is represented with several well-annotated genomes and/or proteomes (*Figure 1—figure supplement 1*, *Supplementary file 1*–Supplementary Table 3). Importantly, sequence data from 19 cnidarian species, including four sequenced genomes and five transcriptomes with CEGMA scores above 70% (*Supplementary file 1*–Supplementary Table 2), allow the crucial distinction of orthogroups with cni-darian participation from bilaterian-specific orthogroups without cnidarian contribution, a serious problem of existing databases (*Table 1*).

While other orthology databases might surpass the BigWenDB in species number, this is often due to the integration of many non-metazoan and prokaryotic species (*Table 1*). Still, the total sequence content of other databases is small enough to be handled by a MySQL engine (see http://www.orthodb.org/v9.1/download/README.MySQL.txt; www.orthomcl.org) because it is restricted to predicted and annotated protein sequences. To accomplish processing of the large amount of sequence data from 25 genomic ORF sets, we developed an R-based version of the OrthoMCL pipe-line (*Li et al., 2003*). It reproduces the results of the original pipeline meticulously (*Supplementary file 1*–Supplementary Table 4) and is capable of processing at least 125 million sequences with current computer hardware, considerably extending the limit imposed by conven-tional MySQL usage. In view of the ongoing increase in sequence data, the R-based version of OrthoMCL may prove valuable for generating large and comprehensive orthology datasets in the future.

Importantly, scaling up the orthology engine to handle larger datasets did not come at the expense of clustering quality. Rather, the combination of a comprehensive dataset and a scalable orthology prediction tool turned out as beneficial, challenging an early study that found a high false-positive rate when testing OrthoMCL on a small and taxonomically restricted dataset (*Chen et al., 2007*). This advance of our approach is further demonstrated by correct orthology inference rates that surpass those previously obtained in the orthobench comparisons (*Trachana et al., 2011*; *Supplementary file 3*).

### Reciprocal HMM-HMM comparisons for improving orthogroup completeness

Despite the existence of many orthology detection methods (*Tekaia, 2016*), current tools do not evaluate orthogroup composition after clustering. In contrast, we implemented filtering steps to first

identify widely distributed bilaterian-specific orthogroups. We then applied to the resulting orthogroups extensive procedures for quality control and error correction, taking into account the taxonomic composition of orthogroups and their best hits in HMM-HMM searches. In this context, we developed a new reciprocal HMM-HMM comparison step to evaluate orthogroup completeness because reliable orthogroups are a prerequisite for inferring the evolutionary age of the corresponding gene (*Supplementary file 1*–Supplementary Table 7). Although HMMs generated from orthogroup alignments can be uninformative outside conserved regions, they capture important amino acid positions and their spacing and variability, and therefore the individual profile of an orthogroup even within common functional domains such as zinc fingers (*Supplementary file 1*–Supplementary Figure 2). Indeed, we observed several instances where HMM-HMM comparisons improved results and affected conclusions, demonstrating the value of this novel step (*Supplementary file 1*–Supplementary Table 13, *Supplementary file 1*–Supplementary Table 14, *Supplementary file 1*–Supplementary Table 16, *Supplementary file 1*–Supplementary Table 19, *Supplementary file 1*–Supplementary Table 21, *Supplementary file 1*–Supplementary Table 22, *Supplementary file 1*–Supplementary Table 23, *Supplementary file 1*–Supplementary Table 24).

In particular, we employed highly sensitive HMM-HMM comparisons to minimise errors caused by low protein traceability, the limitation of the BLAST algorithm to detect orthologous genes in distantly related organisms (*Jain et al., 2019*; *Weisman et al., 2020*). This strategy led to the removal of 68 false-positive orthogroups from an initial set of 431 bilaterian-specific orthogroups because they displayed reciprocal best-hit relationships to non-bilaterian orthogroups, indicating a more ancient origin (see Appendix 1: Identification of bilaterian-specific genes). In addition, the broad coverage of bilaterians and non-bilaterians and the evaluation of orthogroup composition by filtering rules minimises errors that may be caused by the low traceability of specific genes or by single taxa with particularly high evolutionary rates.

## Limitations of our orthology clustering pipeline

Our methods for error correction facilitate the detection of reliable lineage-specific gene sets and may serve as a future standard. However, developing software that can automatically detect such patterns and combine/split orthogroups in awareness of the underlying phylogeny would further improve orthogroup assignments. That lineage-specific genes exist and can directly change an animal's phenotype to gain access to new ecological niches has been shown recently, illustrating the importance of these genes and the need for their identification (*Dunwell et al., 2017*; *Santos et al., 2017*; *Luis Villanueva-Cañas et al., 2017*).

Although we obtained a robust set of 157 genes that evolved in the bilaterian ancestor or, more specifically, in the ancestor of protostomes and deuterostomes (Nephrozoa) (*Jondelius et al., 2002*), by design our study is limited to protein coding sequences. It will therefore miss the possible involvement of RNA genes in bilaterian evolution, including miRNAs (micro RNAs) and lncRNAs (long non-coding RNAs), as suggested by *Prochnik et al., 2007*. It will further fail to detect changes in cis-regulatory regions and structural alterations or epigenetic changes, additional factors that affect evolutionary processes (*Carroll, 1995*; *Prud'homme et al., 2006*; *Klironomos et al., 2013*; *Feulner and De-Kayne, 2017*). Despite these limitations, our study successfully corroborated the bilaterian origin of several previously known bilaterian-specific genes, such as the chromatin organiser CTCF (*Heger et al., 2012*), the left-right determination factor Nodal (*Grande et al., 2014*), and central Hox genes (*Finnerty and Martindale, 1999*; *Hueber et al., 2013*).

## Challenges in reconciling orthogroups and phylogenetic trees

Orthology clustering is a distinct method from phylogenetic tree building, and when we used phylogenetic analyses to validate orthogroup composition, we experienced difficulties in reconciling the two approaches.

Firstly, we do consistently obtain high branch support for bilaterian-specific orthogroups as discrete clades. Yet within orthogroups, phylogenetic resolution was often weak, with low branch support and gene tree–species tree discordance. However, tree discordance in itself does not argue against orthology because phylogenies suffer from various problems, such as the inclusion of problematic sequences, little phylogenetic information, or—in our case—the presence of short ORF fragments (*Aguileta et al., 2008*; *Som, 2015*). While our ORF data aid the recognition of distinct

orthogroups by avoiding systemic annotation errors from external databases and by providing essential taxonomic coverage, these sequences do not represent full-length proteins and may curtail within-orthogroup resolving power.

In addition, in several cases we obtained tree topologies that could imply orthogroup origin in the metazoan ancestor rather than a later, bilaterian origin (*Figure 5—figure supplement 3*, *Figure 5—figure supplement 4*, *Figure 7—figure supplement 1*). One major confounding factor for correct tree reconstruction is heterotachy: a non-constant rate of evolution among different lineages (*Lopez et al., 2002*; *Wu and Susko, 2011*; *Jayaswal et al., 2014*). Importantly, heterotachy is often observed along the branches originating from a gene duplication event (*Kondrashov et al., 2002*; *Conant and Wagner, 2003*; *He and Zhang, 2005*; *Steinke et al., 2006*). Accelerated evolution in bilaterian-specific duplicates could therefore explain the observed tree topologies and the discrepancy between trees and clustering results. In contrast, the alternative interpretation of metazoan orthogroup origins would require that one of the two duplicates was secondarily lost in the stem lineage of sponges, ctenophores, placozoans, and cnidarians because of its absence in all available samples from these phyla. Gene loss is increasingly recognised as a widespread and important evolutionary mechanism (*Sharma et al., 2018*; *Hecker et al., 2019*; *Thomas et al., 2020*). However, the loss of a number of genes in the stem lineages of four independent phyla would imply strong selective pressure against their presence in non-bilaterian lineages, creating an aspect of deep evolution worthwhile of future exploration.

## A robust associaton between bilaterian-specific genes and key morphological features

Several morphological features are widely considered key bilaterian innovations: (i) a third germ layer, the mesoderm; (ii) a complex bilateral nervous system; (iii) a Hox gene cluster with at least seven anterior, posterior, and central Hox genes; (iv) a through gut; (v) an excretory system; (vi) the possession of many different cell types; and (vii) bilateral symmetry (*Baguñà et al., 2008* and references therein). It was unknown so far whether, and if so which, genetic factors contributed to the emergence of these innovations. From the results presented here, we conclude that a considerable fraction of the identified 157 bilaterian-specific genes is associated with the origin of characteristic bilaterian traits. Although correlations cannot prove a causal relationship, in the absence of ancestral genetic information our inferences from extant animals offer a fruitful approach. Here, we elaborate on several instances where the origin of proteins and bilaterian traits appear to coincide.

For example, a large portion of the 157 genes is involved in nervous system development and/or maintenance (*Supplementary file 4*). Several factors in this category provide functionalities absent from non-bilaterian metazoans, such as the long-range control of behaviour and physiological state through an expanded repertoire of GPCRs (*Supplementary file 1*–Supplementary Table 17, *Supplementary file 1*–Supplementary Table 18), a midline repulsion mechanism for the establishment of a bilateral nervous system (Robo-Slit; *Figure 6—figure supplement 3*; *Supplementary file 1*–Supplementary Table 19, *Supplementary file 1*–Supplementary Table 21), or mechanisms for sophisticated axon guidance and synaptic plasticity (neurotrophin signalling system; *Figure 7*; *Supplementary file 1*–Supplementary Table 22, *Supplementary file 1*–Supplementary Table 23, *Supplementary file 1*–Supplementary Table 24). These findings are consistent with the convergent evolution of muscle and nerve cells in ctenophores (*Moroz et al., 2014*) and suggest that bilaterians have a common genetic basis for nervous system patterning despite the recently proposed scenario of convergent evolution of bilaterian nerve cords (*Martín-Durán et al., 2018*). The importance of the nervous-system-related category of bilaterian-specific genes is further underscored by the identification of various transcription factors with a well supported role in nervous system development across phyla, for example the Prospero homeobox protein, the Achaete-scute homologue 2, or the neuronal PAS domain-containing protein 4 (*Supplementary file 1*–Supplementary Table 10, *Supplementary file 4*). Further, three transmembrane proteins with expression in the nervous system, but unknown function, provide the opportunity to characterise novel factors with nervous system-related function (*Supplementary file 1*–Supplementary Table 11). Together, the factors we found in this category provide fundamental features of bilaterian nervous systems, and their evolutionary origin in the bilaterian ancestor is compatible with observable changes in nervous system development and architecture.

An unexpectedly high number of bilaterian-specific genes has transcription factor activity (*Figure 3B*; *Figure 2*). As noted above, these factors are often equipped with multiple $C_2H_2$ zinc finger domains (*Figure 2—figure supplement 1*; *Supplementary file 1*–Supplementary Table 8). Apart from so far uncharacterised proteins, which include ZF64B_HUMAN or ZN236_HUMAN, the expression and developmental role of bilaterian-specific zinc finger proteins is compatible with prominent functions during early development, such as imaginal disc development (Rotund; *St Pierre et al., 2002*), modulation of TGF-β signalling (Schnurri; *Yao et al., 2006*), nephron patterning (Evi1, odd-skipped related 1; *Mead et al., 2005*; *Dressler, 2006*; *Tena et al., 2007*; *Li et al., 2014*), or the differentiation of cardiac precursor cells at the ventral midline (Castor; *Christine and Conlon, 2008*). Importantly, the identified transcription factors with homeobox or bHLH domain are involved in the specification of several bilaterian tissues, the mesoderm (MyoD, PRRX1_HUMAN, BHE22_HUMAN), the nervous system (Prospero homeobox protein 2, Achaete-scute homologues 2, FER3L_HUMAN, NPAS4_HUMAN, BHE22_HUMAN, BUN1_DROME), or the intestine (ISX_HUMAN), consistent with a role in the evolution of these characteristic bilaterian traits .

A contiguous cluster of at least seven Hox genes is an ancestral bilaterian feature (*Baguñà et al., 2008*). A prerequisite for its formation is the existence of anterior, central, and posterior Hox genes. Our results confirm previous findings that placed the origin of central Hox genes to the bilaterian ancestor (*Supplementary file 1*–Supplementary Table 10), in contrast to evolutionarily older anterior and posterior Hox genes (*Finnerty and Martindale, 1999*; *Hueber et al., 2013*). Importantly, Hox gene expression is regulated in part by the chromatin organiser CTCF (*Rousseau et al., 2014*; *Narendra et al., 2015*), another bilaterian-specific protein (*Heger et al., 2012*; *Supplementary file 1*–Supplementary Table 8; *Supplementary file 4*). As outlined elsewhere, the evolution of CTCF—and other poly-zinc finger proteins—could have provided a mechanism for the creation and regulation of bilaterian Hox gene clusters, once central Hox genes had been added to the repertoire (*Heger et al., 2012*).

The emergence of the mesoderm as a third germ layer is one of the most characteristic morphological innovations of bilaterian animals. In contrast to previous work, our findings suggest that several genes and gene networks which provide regulatory inputs to mesodermal patterning arose in the bilaterian ancestor. Specifically, we identified orthologues of all Nodal pathway members across bilaterians, but not outside this clade (*Figure 5—figure supplement 1*, *Figure 5—figure supplement 2*, *Figure 5—figure supplement 3*, *Figure 5—figure supplement 4*; *Supplementary file 1*–Supplementary Table 14, *Supplementary file 1*–Supplementary Table 16). The robust bilaterian-specific distribution of these genes, derived from orthology clustering and HMM-HMM searches, implies that the entire Nodal pathway—and its roles in mesoderm specification and left-right asymmetry—is a bilaterian novelty (*Figure 5*). Although a reasonable speculation, this is currently not supported for all pathway members by phylogenetic analyses and needs to be tested more thoroughly in the future. Together with the bilaterian specificity of additional modulators and effectors of Nodal and/or TGF-β signalling (BAMBI_HUMAN, VWC2_HUMAN, MECOM_HUMAN, Q24605_DROME; *Supplementary file 4*), these findings suggest that significant changes in TGF-β signalling occurred in the bilaterian ancestor. In addition to the Nodal pathway, several other genes with key roles in mesoderm formation also originated in the bilaterian ancestor, among them the master regulator of muscle cell specification, MyoD, and the Paired mesoderm homeobox protein 1 (PRRX1_HUMAN; *Supplementary file 1*–Supplementary Table 10) which regulates the formation of preskeletal condensations from undifferentiated mesenchyme during mouse skeletogenesis (*Martin et al., 1995*). Taken together, we identified multiple genetic factors essential for the differentiation of mesoderm and mesodermal tissues in bilaterians.

In conclusion, we demonstrate that a considerable number of genes has a bilaterian-specific distribution and probably originated in the bilaterian ancestor. While the function of some of these genes is unknown, many of them participate in the formation of key morphological innovations in extant bilaterians, implying that the evolution of specific genes contributed to the formation of bilaterian body plans.

# Materials and methods

## Sequence collection and database construction

The sequence repertory for this study was assembled from three parts. Genomic and transcriptomic sequences were collected from the sources listed in *Supplementary file 1*–Supplementary Table 1, *Supplementary file 1*–Supplementary Table 3, *Supplementary file 2*. As third component, selected sequences were downloaded from the NCBI non-redundant protein database.

The 25 genomic sequences were first screened for repetitive sequence content using Repeat-Masker V4.0.5 (http://repeatmasker.org) with default parameters. The resulting contigs/scaffolds were translated into six ORFs using the Emboss tool "getorf" (*Rice et al., 2000*), with a minimum ORF length of 25 AA. Sequences containing strings of "X" characters, a result of translating sequencing gaps and masked repeats, were treated differentially to retain as much information as possible. Sequences with $\geq$9 "X" in a row were split. After removing the Xs, each flanking region $\geq$35 valid amino acids was kept and given a new identifier while smaller flanking regions were discarded. These measures decreased sequence count by 46.8%, from 324,788,561 to 172,606,165 ORFs. To further reduce the amount of ORFs, we blasted them against a custom database of opisthokont sequences. This database contained all sequences of opisthokont origin as extracted from the non-redundant protein database at GenBank, release 198 from 21 October 2013 (2,695,641 sequences). We kept ORFs with a BLAST expectation value <10 against this database and thus rejected ORFs that have no detectable similarity to the protein repertoire of opisthokonts. In a final step, we used CD-HIT (*Li and Godzik, 2006*) with default parameters and 90% identity threshold to remove redundancy. These steps reduced the number of sequences significantly, from initially 324,788,561 to 109,567,344 genomic ORFs.

To fill in the gaps of public sequence repositories and extend coverage, we collected transcriptome data of poorly represented animal groups (*Supplementary file 1*–Supplementary Table 1, *Supplementary file 2*). Downloaded transcriptomes were first assayed for completeness using the CEGMA (Core Eukaryotic Genes Mapping Approach) pipeline which reports the coverage of 248 ultra-conserved core eukaryotic genes present in a dataset (*Parra et al., 2007*). On the basis of CEGMA completeness and phylogenetic placement, we selected transcriptomes of 64 species for the dataset. Their average transcriptome completeness according to CEGMA was 60.8%, with several bilaterian and non-bilaterian species exceeding 90% (*Supplementary file 1*–Supplementary Table 2). As described for genomes, transcriptomes were then translated into six ORFs. We kept the three longest ORFs for each transcriptome contig, removed Xs, and obtained 11,768,516 transcriptome protein sequences in total (*Table 2*).

To provide a backbone of published and annotated protein sequences for the genomic and transcriptomic ORFs, we filtered the NCBI non-redundant protein database and kept 2.9 million protein sequences from 204 opisthokont species that had >8000 sequence entries each. Extraction of opisthokont sequences was guided by NCBI taxonomy.

As the combination of sequences from three sources again introduced redundancy, we clustered the final dataset with 90% identity threshold. In a last pre-processing step, we changed the headers of all sequences to obey a consistent naming scheme. It includes the NCBI taxon identifier and a unique sequence ID that allows to distinguish between NCBI-, ORF-, and transcriptome-derived sequences. The final dataset used for this analysis contained 124,031,501 sequences.

## Orthology pipeline and clustering

For orthology clustering, we employed the OrthoMCL pipeline (*Li et al., 2003*). It utilises a graph-based clustering approach for the generation of orthologous groups on the basis of normalised BLAST similarity measurements between sequence pairs. To enable the processing of our large dataset, we ported to the statistical programming environment R (https://www.r-project.org/) all steps of the original OrthoMCL pipeline that require interaction with a MySQL database. In this way, loading of the database and inference of orthology tables is limited only by the size of the computer's main memory, not by the speed and additional memory requirements of the underlying MySQL engine, as in the original implementation. By dividing the computation of orthology tables into an appropriate number of steps, our entire dataset could be processed on a compute server with 250 GB memory. Importantly, the R version of OrthoMCL accurately reproduces all steps of the original pipeline

(*Supplementary file 1*–Supplementary Table 4). The collection of scripts for the R version of OrthoMCL is available at https://github.com/prheger/BigWenDB (*Heger, 2020*; copy archived at https://github.com/elifesciences-publications/BigWenDB).

## HMM-HMM searches and database

We extracted from the BigWenDB sequence collection the individual sequences belonging to each of the 824,605 orthologues groups and calculated 824,605 corresponding multiple sequence alignments using default parameters of the MAFFT v7.304b "einsi" algorithm (*Katoh et al., 2005*). After converting the alignments into hhm format (hhsearch format for hidden Markov models) with the command "hhmake" and default parameters, we concatenated them to a database that can be searched by hhsearch (parameters in addition to default: "-nodssp -nopred -dbstrlen 100"), according to *Söding, 2005*. We precomputed HMM-HMM search results for about 20% of orthogroups and issued missing searches on demand. Reciprocal best hit relationships were analysed using custom scripts.

## Quality control of clustering and the bilaterian-specific gene set

Quality control of clustering results and the bilaterian-specific gene set was carried out as described in Appendix 1, sections "Cluster evaluation and quality control" and "Identification of bilaterian-specific genes".

## Statistical tests for the enrichment of transcription factors

To test whether the bilaterian-specific gene set of 157 orthogroups is enriched for transcription factors, we downloaded as control the human proteome with 20,205 protein sequences from ftp://ftp.uniprot.org/pub/databases/uniprot/current_release/knowledgebase/reference_proteomes/Eukaryota/ and predicted transcription factors in this dataset using the PfamScan software (ftp://ftp.ebi.ac.uk/pub/databases/Pfam/Tools/) with E-value cutoff = $5 \times 10^{-05}$. We then determined the abundance of 10 prevalent DNA-binding domains in the dataset: "Basic; bZIP_2; HLH; HNF-1_N; Homeobox; Hox9_act; HPD; SOBP; THAP; zf-". Corresponding domains were identified in 1,756 of the 20,205 human reference proteins. We then randomly selected $10 \times 157$ genes from the reference set and specified the number of transcription factors (proteins with the above mentioned domains) in the obtained subsets. While the average number of transcription factors in the 10 control sets was $12.8 \pm 4.44$, the equally sized bilaterian-specific gene set (157 orthogroups) had 37 transcription factors. Modelling a normal distribution from the obtained mean and standard deviation yielded a $p$-value of $2.512e^{-08}$ for the transcription factor content in bilaterian-specific genes (using the R function "pnorm"). Likewise, a Pearson's $\tilde{\chi}^2$ test with the corresponding data matrix (1,765:20,205; 37:157), using the R function "chisq.test", yielded a $p$-value of $3.805e^{-08}$. Finally, under the assumption of a binomial distribution (R function "pbinom") and given that there are 1,756 transcription factors in 20,205 human proteins, the probability that we obtain 36 or more transcription factors when drawing 157 random proteins is $p < 1.841e^{-08}$.

## Poly-Zinc finger scan across Opisthokonta

We downloaded the proteomes of 7 ecdysozoan, 5 lophotrochozoan, 12 deuterostomian, and 4 non-bilaterian species from http://www.uniprot.org/proteomes. On average, each proteome consisted of 28,772 sequences. We scanned all protein sequences for the presence of protein domains using the PfamScan software (ftp://ftp.ebi.ac.uk/pub/databases/Pfam/Tools/) with E-value cutoff = $5 \times 10^{-05}$ and Pfam database version 31.0. Using command line tools, we identified $C_2H_2$ zinc finger proteins in the PfamScan output and counted for every proteome the number of proteins with six or more zinc finger domains. The resulting numbers were used to plot *Figure 2—figure supplement 1A,B*.

To determine the number of poly-ZF proteins that originated in the ancestor of opisthokonts, metazoans, and eumetazoans, we first extracted from the clustering results orthogroups specific for these lineages. The filtering criteria for selecting opisthokont-specific orthogroups were: Fungi $\geq$ 20 species, Metazoa $\geq$ 40 species, Bilateria $\geq$ 30 species and yielded 2,928 orthogroups of ancient origin. The filtering criteria for selecting metazoan-specific orthogroups were identical, except that no fungi were allowed, and yielded 2,615 metazoan-specific orthogroups. For eumetazoan-specific

orthogroups we required the presence of at least 30 bilaterian and 3 cnidarian species, with not more than 2 ctenophore species allowed (according to NCBI taxonomy, both ctenophores and cnidarians misleadingly belong to eumetazoans). Applying these conditions, we obtained 283 eumetazoan-specific orthogroups. Next, we extracted the longest sequence of each opisthokont-, metazoan-, and eumetazoan-specific orthogroup and scanned it with PfamScan (E-value cutoff = $5 \times 10^{-05}$). Finally, we counted the number of poly-ZF sequences with at least six domains for each node and mapped the numbers to a phylogeny. Note that this "simple" filtering strategy (Bilateria: $\geq$ 30 species) would return 662 bilaterian-specific orthogroups, considerably more than the 157 error-corrected orthogroups in the final dataset. The strategy therefore possibly overestimates the number of poly-ZF proteins at the three ancient nodes.

## Determining orthogroup ancestors

To determine the ancestor of the species combined in a given orthogroup, we wrote a custom Perl script that extracts the taxonomic identifiers of each sequence and then determines the last common ancestor of all represented species on the basis of NCBI taxonomy and lineage information (ftp:// ftp.ncbi.nlm.nih.gov/pub/taxonomy/). The script generates output that can be parsed and filtered using command line utilities. It is part of the collection of R scripts at https://github.com/prheger/ BigWenDB.

## Protein domain scans and gene ontology analysis

We applied strict filtering rules to extract bilaterian-, vertebrate-, and arthropod-specific genes from the Markov clustering results (rule for bilaterian-specific orthogroups: deuterostomes $\geq$ 7, lophotrochozoans $\geq$ 4 or 0, ecdysozoans $\geq$ 4 or 0; for arthropod-specific orthogroups: chelicerates $\geq$ 2, crustaceans $\geq$ 0, myriapods $\geq$ 1, insects $\geq$ 5; for vertebrate-specific orthogroups: $\geq$40 of 53 gnathostome species). From each lineage-specific orthogroup obtained by filtering, we extracted the longest sequence and scanned it with PfamScan Version 1.5 (*Punta et al., 2012*) (available at ftp://ftp.ebi.ac.uk/pub/databases/Pfam/Tools/) at an E-value cutoff of $e^{-05}$ for the presence of protein domains as classified in PFAM database release 30.0 (release date: 06/16).

To associate the identified protein domains with gene ontology (GO) terms, we utilised the Pfam2GO list at http://geneontology.org/external2go/pfam2go and extracted relevant terms using command line tools. Typically, only a subset of domains was linked to GO terms. We finally created a list with the relative number of identified protein domains and associated gene ontology terms and visualised this list as a word cloud at www.wortwolken.com.

## Multiple sequence alignment and phylogenetic analysis

Multiple sequence alignments required for the HMM-HMM database and phylogenetic analyses were carried out using the MAFFT v7.304b "einsi" algorithm with default parameters (*Katoh et al., 2005*). Large alignments (>200 sequences) were computed using MAFFT v7.304b with high-speed parameters. For phylogeny, we added ingroup and outgroup sequences from the clustered orthogroup sets or from public repositories, as appropriate, and manually removed indels and unalignable regions from the data prior to analysis. In some cases, for example for Lefty, we generated a hidden Markov model of an orthogroup alignment and searched additional transcriptomic datasets not represented in the BigWenDB for potential orthologues. Phylogenetic trees were computed under the maximum likelihood criterion, using IQ-TREE v1.6.10 (*Nguyen et al., 2015*) with ModelFinder for fast and accurate model selection (*Kalyaanamoorthy et al., 2017*), ultrafast bootstrap approximation and optimisation (1000 replicates) (*Minh et al., 2013*), and Shimodaira-Hasegawa-like approximate likelihood ratio test (SH-aLRT) (command line parameters: "-bb 1000 -alrt 1000 -bnni"). Resulting trees were edited with FigTree v1.4.3 (http://tree.bio.ed.ac.uk/software/figtree/) and Affinity Designer Version 1.72 (https://affinity.serif.com).

## Prediction of protein structure

After constructing multiple sequence alignments from cnidarian and bilaterian Robo proteins, we identified the transmembrane region (corresponding to sequence "AFIAGIGAACWIILMVFSIWL" in ROBO1_HUMAN) and generated two subsequences overlapping at this feature. One subsequence spanned the extracellular part of the protein plus the transmembrane domain, the other spanned

the transmembrane domain plus the cytoplasmic part. We generated the two fragments for seven exemplary Robo proteins, for the deuterostomes *Homo sapiens* and *Strongylocentrotus purpuratus*, the lophotrochozoan *Lingula anatina*, the ecdysozoans *Drosophila melanogaster* and *Trichinella pseudospiralis*, and the two cnidarians *Hydra vulgaris* and *Stylophora pistillata*. All fragments were uploaded to the Phyre2 web interface (http://www.sbg.bio.ic.ac.uk/phyre2/html/page.cgi?id=index; *Kelley et al., 2015*) and analysed with modelling mode "intensive" (complete modelling using multiple templates and ab initio techniques).

## Identification of metazoan-specific genes

To obtain a list of genes with metazoan origin, we first blasted 20,205 human genes obtained from uniprot.org against the BigWen database and obtained BLAST hits for 19,322 genes. To reliably map the UniProt queries to orthogroups, we selected queries that had a BLAST hit with high identity (>95%) over at least 100 amino acids. For proteins fulfilling these criteria, we extracted the corresponding orthogroup ID and ancestor, taking into account only orthogroups with at least 75 species to ensure broad sampling. After removing redundancy, we obtained 797 distinct orthogroups of metazoan origin whose human orthologues were used for the stringDB PPI network analysis. A conceptually similar study obtained 1,189 novel metazoan-specific homology groups, which is in reasonable agreement with our result when considering the differences in methodology and datasets (*Paps and Holland, 2018*).

## Protein-protein interaction network analyses

Protein interaction data were obtained from the STRING database v11.0 of known and predicted protein-protein interactions (PPI; https://string-db.org; *Szklarczyk et al., 2017*). To construct PPI networks, we first identified the appropriate human orthologues of bilaterian-specific and metazoan-specific orthogroups. We obtained 150 human orthologue IDs for the 157 bilaterian-specific orthogroups and 797 human orthologue IDs for 797 metazoan-specific orthogroups (collected as described above). We uploaded these protein IDs to the STRING browser interface and generated three separate PPI networks, one for bilaterian-specific proteins (B), one for metazoan-specific proteins (M), and a combined network for both taxonomic groups (B + M). The average local clustering coefficients and PPI enrichment $p$-values we report are based on analyses with default settings, where all evidence types were considered. Further statistical analyses were conducted for the B + M full network and the B + M Nodal-Lefty subnetwork, the latter being defined by the core five bilaterian-specific proteins (Nodal, Lefty, FoxH1, Eomes, and EGF-CFC) and their interaction partners. From the complete list of pairwise protein-protein interactions in the B + M network, data were extracted for the numbers of B – B, M – M, and B – M interactions and assessed by a $\tilde{\chi}^2$ test. Additional calculations were made per protein for the total number of interactions and for the proportion of interactions that involve a bilaterian-specific partner. Boxplots for these values display the median, and whiskers represent 1.5× the value of the Q3 (upper) or Q2 (lower) quartile range, with outliers omitted for clarity. Statistical tests involved $\tilde{\chi}^2$ tests (https://www.socscistatistics.com/tests/chi-square/default2.aspx, accessed 26 August 2019) and non-parametric comparisons in multigroup (Kruskal-Wallis) and pairwise (Mann-Whitney U) assessments as reported, calculated in R version 3.4.0 and from the Python library scipy.stats (function: mannwhitneyu).

## Data access

The R version of OrthoMCL and a script for inferring orthogroup ancestors are available at https://github.com/prheger/BigWenDB. The sequence dataset used to build the BigWenDB and the final clustering results are available at https://doi.org/10.5061/dryad.4qf7168. Several Supplementary Files with original data and Supplementary Tables are linked to this paper at elifesciences.org.

## Acknowledgements

This research was supported by grants from the German Research Foundation to TW (CRC 680 and CRC 1211) and to KAP (CRC 680). BLAST searches were computed on CHEOPS, the Cologne High Efficiency Operating Platform for Science of the University of Cologne, and on JuRoPA (Jülich Research on Petaflop Architectures), a High Performance Computing Platform of the Jülich

Supercomputing Centre, Germany. We thank Robert Fürst for programming help, Kay Hofmann for help with protein structure analysis, Richard Stancliffe for scripting and statistical support, Maria Thieser for help with transcriptome processing, and Olav Zimmermann for the cooperation with the Jülich Supercomputing Centre. Special thanks to countless researchers and institutions for sharing sequence data.

## Additional information

### Funding

| Funder | Grant reference number | Author |
|---|---|---|
| Deutsche Forschungsgemeinschaft | CRC 680 | Kristen A Panfilio Thomas Wiehe |
| Deutsche Forschungsgemeinschaft | CRC 1211 | Thomas Wiehe |

The funders had no role in study design, data collection and interpretation, or the decision to submit the work for publication.

### Author contributions

Peter Heger, Conceptualization, Data curation, Software, Supervision, Validation, Investigation, Visualization, Methodology, Project administration; Wen Zheng, Data curation, Software, Validation, Investigation, Methodology; Anna Rottmann, Software, Investigation; Kristen A Panfilio, Resources, Supervision, Funding acquisition, Validation, Investigation; Thomas Wiehe, Conceptualization, Resources, Supervision, Funding acquisition

### Author ORCIDs

Peter Heger https://orcid.org/0000-0003-2583-2981
Kristen A Panfilio https://orcid.org/0000-0002-6417-251X
Thomas Wiehe https://orcid.org/0000-0002-8932-2772

### Decision letter and Author response

Decision letter https://doi.org/10.7554/eLife.45530.sa1
Author response https://doi.org/10.7554/eLife.45530.sa2

## Additional files

### Supplementary files

- Supplementary file 1. Supplementary Tables 1–26 and Supplementary Figures 1 and 2.
- Supplementary file 2. Download location for transcriptome data used in this study.
- Supplementary file 3. Comparison between Orthobench and BigWenDB clustering results.
- Supplementary file 4. List of high-confidence bilaterian-specific orthogroups.
- Transparent reporting form

### Data availability

Accession numbers and/or URLs for previously published transcriptome datasets are listed in Supplementary File 3. Download links for previously published genomic sequences are listed in Supplementary File 1-Supplementary Table S7. Orthology datasets generated in this study have been deposited to Dryad, under the URL https://doi.org/10.5061/dryad.4qf7168.

The following dataset was generated:

| Author(s) | Year | Dataset title | Dataset URL | Database and Identifier |
|---|---|---|---|---|
| Heger P, Zheng W, | 2019 | Data from: The genetic factors of | https://doi.org/10.5061/ | Dryad Digital |

| Rottmann A, Panfilio K, Wiehe T | bilaterian evolution | dryad.4qf7168 | Repository, 10.5061/dryad.4qf7168 |
|---|---|---|---|

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

# Appendix 1

## Orthology pipeline and clustering

To generate clusters of orthologous proteins from the collected sequence data, we used the OrthoMCL pipeline (*Li et al., 2003*). OrthoMCL is a graph-based method for orthologue group identification that represents sequences as nodes and their similarities as weighted edges. A normalization step adjusts initial similarity scores to reflect species distance and ensures that edge weights for sequence pairs are comparable between different genomes. Finally, the Markov cluster algorithm (*van Dongen, 2000*) performs random walks on the normalised graph by simulating transition probabilities of sequences to other nodes, thereby revealing an underlying cluster structure. To create the BLAST similarity table required by OrthoMCL, we performed all-vs-all BLAST searches with 124 million sequences (with default BLAST parameters, except "-outfmt 6"; BLAST version 2.2.28). Roughly one million CPU hours were necessary for this task, running hundreds of jobs in parallel on a high performance computing platform. Merging the individual output files, we obtained a similarity score table of ~500 GB, containing roughly 6 billion BLAST hits (see *Supplementary file 1*–Supplementary Table 5). In the original implementation, OrthoMCL loads the BLAST output table into a MySQL database and performs subsequent computations within the relational database. Because of its size, we could not load the BLAST output table into a physical MySQL database. We therefore ported all MySQL processes to the statistical computing environment R to execute them in computer memory. Test experiments, carried out in parallel with our R implementation and the original software, produced identical results, demonstrating that the R version of OrthoMCL accurately reproduces the outcome of the standard pipeline (*Supplementary file 1*–Supplementary Table 4). After obtaining the final table with adjusted pairwise distance information in R, we used Markov clustering (*van Dongen, 2000*), as in the original protocol, to combine sequences to orthologous groups.

Depending on the origin of compared sequences, OrthoMCL creates three orthologue tables: a table with reciprocal relationships of sequences between different species (orthologue table), a table of within-species relationships (in-paralogues), and a table of co-orthologues with protein pairs that are connected through orthology and in-paralogy. Of 124 million gathered sequences, 122 million had at least one BLAST hit in the database, giving rise to a collection of 6 billion BLAST pairs as raw material for orthology clustering. The OrthoMCL pipeline retained 35 million of these sequences in 806 million pairs of the three orthology tables. Thus, 28.8% of the sequences had enough similarity with other sequences to participate in orthology group construction whereas the majority of input sequences were so remotely related to other sequences that our pipeline could not merge them with a cluster. As expected, artificially generated ORFs represented by far the largest portion of the non-clustered sequences (91.3%).

As we observed that a large in-paralogue table (5.8× larger than the orthologue plus co-orthologue tables for the final dataset) negatively affected the accuracy of the clustering process, we omitted this table in subsequent trials. In the final MCL run, we obtained 824,605 orthologous groups with 6,743,519 distinct sequences derived from 118,499,524 protein pairs (BLAST hits) of the orthologue and co-orthologue tables. Discarding the large in-paralogue table led to a drop in the percentage of clustered sequences from 28.8% to 5.5%, indicating that a considerable amount of orthogroups in the larger dataset consisted of paralogues (*Supplementary file 1*–Supplementary Table 5).

To investigate the properties of orthogroups as old as bilaterians or older, we plotted for these orthogroups the number of species against their proportion of bilaterians (*Appendix 1—figure 1*). Position and abundance of many data points in the resulting plot are a consequence of dataset composition. For example, (i) the majority of orthogroups is small, leading to an abundance of solid (because of overlap) data points for small orthogroups (*Appendix 1—figure 1*, left part; *Supplementary file 1*–Supplementary Table 5); (ii) bilaterians and non-bilaterians including fungi are groups roughly equal in size (142 vs. 131 species), preventing that bilaterian sequences exceed a coverage of ~50% in large orthogroups. Similarly, bilaterian content can hardly fall below 40% to 50% in large orthogroups with more than ~175 species, giving rise to an arrowhead shape at the right side of the plot (*Appendix 1—figure 1*). (iii) orthogroups with a bilaterian ancestor have, by definition, a bilaterian content of 100% and are therefore spread as dotted red line on top of the plot that is fading away in orthogroups with more than 100 species; (iv) orthogroups with metazoan

and eumetazoan ancestor (green and blue) concentrate on the left part of the plot because not more than 33 non-bilaterian metazoans are present in the dataset, restricting orthogroup size. In addition, the low orthogroup density in sectors B2, B3, and C3 suggests that ancient genes, that evolved in the ancestor of eumetazoans or earlier and survived in bilaterians, do not get lost randomly at multiple nodes in the bilaterian tree. Instead, they tend to be maintained across most bilaterian species. It remains to be seen whether this behaviour is specific for bilaterians in this dataset or a general evolutionary pattern.

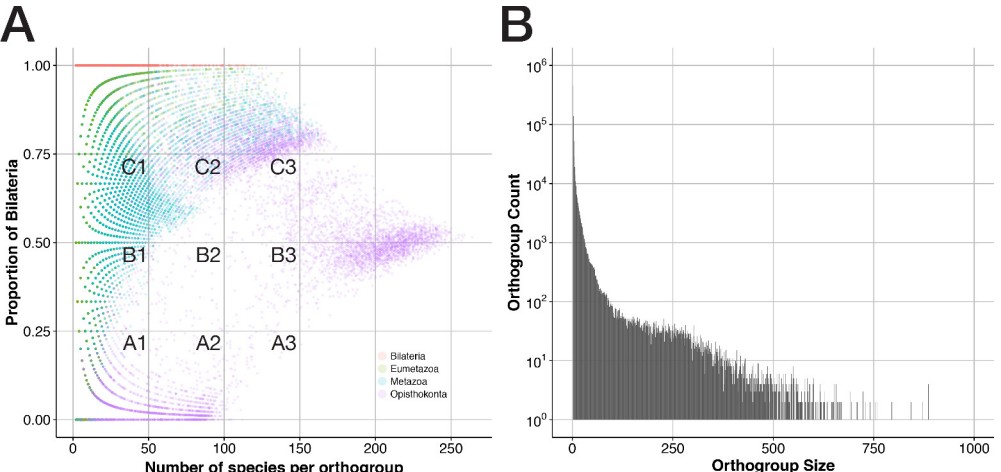

**Appendix 1—figure 1.** General properties of sequence clusters from a bilaterian viewpoint. (**A**) The proportion of bilaterians per orthogroup is shown as a function of orthogroup size (in terms of species number) for 207,285 orthogroups that trace back to the four ancestors Bilateria, Eumetazoa, Metazoa, and Opisthokonta. Dot colours indicate the orthogroup ancestor and are printed with 85% transparency to reveal overlaps. (**B**) Orthogroup count (how often orthogroups of a given size are observed) is displayed as function of orthogroup size (number of sequences present in an orthogroup). 34 orthogroups with more than 1,000 sequences were omitted. Almost all of these sizes occurred only once.

## Cluster evaluation and quality control

In a first approach to verify the accuracy of our clustering results, we employed as an external benchmark a manually curated set of 70 orthologous groups (*Trachana et al., 2011*), the orthobench dataset (http://eggnog.embl.de/orthobench). For the members of every orthobench family, we determined the corresponding BigWenDB sequence ID and the cluster ID (orthogroup ID) to which this sequence was assigned during clustering. We then analysed how the members of a given orthobench family were distributed among orthogroups in the BigWenDB. We performed such comparisons for two MCL inflation parameters ($I = 1.3$ and $I = 1.4$) and two database sizes (full database and database without paralogue table). The clustering with the highest agreement to the expected orthobench outcome was the dataset with inflation parameter $I = 1.3$ and without paralogue table (mcl_ortho-coortho_1.3.7; see *Supplementary file 3*). In this dataset, 46 of 70 protein families were assigned correctly, i.e. in 65.7% of the cases our pipeline combined all members of an orthobench family, as expected, in a single orthogroup. However, BLAST hits that allow correct mapping were not found for all orthobench family members, and some members were mapped to erroneously predicted proteins. In such cases, orthobench members may be linked to an orthogroup different from the rest of the family, leading to the impression that several orthogroups exist for this family. According to our estimates, such mapping errors reduce accuracy by at least 5%, suggesting a correct orthology inference rate above 70% for our dataset. In contrast, only 10% to 48% of reference orthogroups were predicted correctly in the orthobench comparison (*Trachana et al., 2011*), indicating that the representative coverage of our dataset considerably improves orthogroup inference quality.

Evolutionary relationships of homeodomain-containing genes are difficult to trace because of the strong conservation and shortness of the homeodomain (60 AA) (*Irvine et al., 1997*; *Kourakis and*

*Martindale, 2000*). To understand how our study deals with these difficulties, we analysed the composition of orthogroups containing NK (Nirenberg-Kim) homeobox genes. Like Hox and ParaHox gene clusters, the NK cluster is a close association of homeobox genes with crucial roles in animal development. It consists of the six genes tinman, bagpipe, ladybird (early and late), C15, and slouch in *D. melanogaster*. They are all involved in mesodermal patterning (*Kim and Nirenberg, 1989*; *Jagla et al., 2001*). Genomic data from vertebrates and the cephalochordate *Branchiostoma* indicate that the NK cluster is an ancient feature of bilaterians, but has been duplicated and split repeatedly in chordate history, leading to the presence of four dispersed clusters and multiple paralogues of each gene in humans (*Luke et al., 2003*). Several rearrangements have also been observed in the NK cluster of arthropods (*Chan et al., 2015*). In addition, studies of the homeodomain gene complement of sponges and cnidarians revealed that NK cluster genes predate the evolution of bilaterians (*Ryan et al., 2006*; *Larroux et al., 2007*). Given these findings, we can expect that NK homeobox genes from diverse metazoans (sponges, cnidarians, vertebrates, and insects) are each represented in a single orthogroup. Analysing the orthogroups of all *Drosophila* and human NK cluster genes revealed that, indeed, bilaterian and non-bilaterian orthologues of the five NK genes were joined in five corresponding groups (*Supplementary file 1*–Supplementary Table 6). These five orthogroups contained sequences from 81 to 128 (of 142) bilaterian species, including the known *Drosophila* and human NK genes, as well as sponge, cnidarian, and ctenophore sequences. We found placozoan sequences in a single orthogroup, OG_613 (NKX2), suggesting the previously unknown existence of NK class homeobox genes in Placozoa (*Monteiro et al., 2006*). In contrast to other NK genes, *Drosophila* tinman is not located in the group of its vertebrate counterparts NKX2.3/2.5/2.6 (OG_613; *Supplementary file 1*–Supplementary Table 6). It has been shown previously that orthology relationships between tinman and vertebrate NKX2 genes are difficult to establish because of the fast evolving insect tinman genes (*Harvey, 1996*; *Saudemont et al., 2008*). In line with these observations, tinman was assigned to a small orthogroup restricted to endopterygote insects (OG_92160) while other putative NKX2 orthologues from a wide range of arthropods (32/37 species) were combined with vertebrate NKX2 genes in orthogroup OG_613.

Consistency between our method and an independent method would further underline the reliability of inferred orthogroups. We therefore prepared our data for a control run with the orthogroup inference algorithm OrthoFinder that, in contrast to OrthoMCL, takes into account a so far unrecognised gene length bias (*Emms and Kelly, 2015*). However, the number of pairwise BLAST similarity tables, resembling OrthoFinder's input, increases quadratically with the number of species, and so does the amount of required main memory. With 80 species and 6,320 corresponding BLAST tables, approximately 250 GB of memory are occupied, precluding a run with the full dataset (273 species; 74,256 BLAST tables) on current computers. OrthoFinder thus cannot be used to confirm our data until it is adapted to large data sets, in turn illustrating the power of our modified version of the OrthoMCL pipeline.

Taken together, the assessment of clustering quality using a benchmark and a homeobox gene set indicates that orthology prediction in the BigWenDB accurately captures known evolutionary relationships of difficult target genes over large evolutionary distances. We conclude therefore that our cluster results are well suited as raw material for the search of bilaterian-specific genes.

## Identification of bilaterian-specific genes

To infer lineage-specific genes, we determined on the basis of NCBI taxonomy (ftp://ftp.ncbi.nlm.nih.gov/pub/taxonomy/taxdump.tar.gz) the last common ancestor of the species present in all 824,605 orthologous groups of the final clustering. Together with other ancient groups such as Metazoa, Eumetazoa, or Opisthokonta, the taxon Bilateria is among the top ten of taxa with the highest counts (42,946 bilaterian-specific orthogroups; *Supplementary file 1*–Supplementary Table 25). While these counts include all orthologue groups that trace back to a given ancestor, the majority of groups contains only few species (see *Figure 4*, *Appendix 1—figure 1*, *Supplementary file 1*–Supplementary Table 5). To obtain meaningful groups with a broad representation across bilaterians, we required that at least 10% of the species of each bilaterian super-phylum must be present (Ecdysozoa $\geq$ 6, Lophotrochozoa $\geq$ 4, and Deuterostomia $\geq$ 7 species). We included orthogroups with zero ecdysozoans or lophotrochozoans if the count for the two other super-phyla met the 10%

threshold, thereby allowing for the loss of bilaterian-specific genes in ecdysozoans or lophotrochozoans. Following these rationales, we obtained 345 bilaterian-specific groups.

At least four types of error might impair our set of bilaterian-specific orthologous groups: (1) An orthogroup is judged older than bilaterians, but is in fact bilaterian-specific (orthogroup too large), (2) an orthogroup is inferred to be bilaterian-specific, but is in fact older (orthogroup too small), (3) an orthogroup is found to be bilaterian-specific, but is in fact younger (orthogroup too large), (4) an orthogroup is considered younger than bilaterians, but is in fact bilaterian-specific (orthogroup too small).

The presence of several bilaterian sequences and a single sequence from an earlier branching eukaryote would conceal the potential bilaterian ancestry of an orthogroup (type 1 error). We therefore searched for orthologue groups with broad bilaterian representation, according to our above mentioned rules, and up to two outgroup sequences. Of 349 orthogroups satisfying these criteria, the majority (263 or 75.3%) contained as outliers sequences of cnidarian origin, the sister group of bilaterians. To maximise the likelihood of detecting true outliers, we considered only organisms without direct sister group relationship for further analysis and obtained 86 additional bilaterian-specific candidate groups with one or two non-bilaterian/non-cnidarian sequences. As the probability is high that these orthogroups contain phylogenetically unrelated outliers and actually originated in the bilaterian ancestor, we ranked them, together with the 345 previous orthogroups, in a set of 431 bilaterian-specific orthogroups.

Type 2 errors can arise if the MCL algorithm does not combine a group with bilaterian ancestry and a group with related sequences from non-bilaterian species although both groups might represent a single natural orthology group. To identify such errors, we computed for all 824,605 orthogroups multiple sequence alignments and turned them into profile hidden Markov models (HMMs) that describe alignment consensus sequences in a probabilistic way (*Eddy, 1998*). We then assembled a database from the HMMs and searched the two next similar profiles for every bilaterian-specific group using sensitive HMM-HMM alignments (*Söding, 2005*). We devised a new reciprocal HMM-HMM alignment comparison step, analogous to the strategy of reciprocal best BLAST hits (*Tatusov et al., 1997*; *Ward and Moreno-Hagelsieb, 2014*), to discover bidirectional best hit orthogroup pairs prognostic for common descent. To demonstrate the power of this method, we analysed the orthogroup distribution of two example proteins, Sprouty, an inhibitor of FGF signalling, and the insulator protein GAGA factor. We found that the orthogroups of both, *D. melanogaster* Sprouty and *D. melanogaster* GAGA factor, were smaller than anticipated considering their reported phylogenetic distribution (*Matus et al., 2007*; *Heger et al., 2013*). In both cases, the reciprocal best hit strategy allowed us to detect highly similar orthogroups with known Sprouty and GAGA factor orthologues that complemented the original orthogroup. After fusion of query and reciprocal best hit orthogroups, the resulting sequence collections matched the expected phylogenetic coverage (*Supplementary file 1*–Supplementary Table 7). Encouraged by these findings, we examined the 431 bilaterian-specific orthogroups accordingly and excluded orthogroups from the list if they satisfied three criteria: (i) their best or second best HMM-HMM hit modifies the ancestor of the resulting fusion group, (ii) their best or second best hit orthogroup is a reciprocal best hit, and (iii) their best or second best hit orthogroup does not contain more than three bilaterian species. With the last criterion we avoid to eliminate orthogroups whose reciprocal best hit is an ancient orthogroup with wide bilaterian representation, an indicator of homology rather than of orthology. The majority of bilaterian-specific orthogroups (84.2% or 363/431 orthogroups) were not affected by this procedure. Therefore we considered them high-confidence bilaterian-specific orthogroups. On the other hand, 68 bilaterian-specific orthogroups (15.8%) were possibly false positives and may have originated in pre-bilaterian time.

If, for example, several insects and a single sequence from a vertebrate populate an orthogroup, a bilaterian ancestor would be computed for this group although, from a phylogenetic point of view, the single vertebrate sequence is more likely an outlier added to the group erroneously. The filtering rules mentioned above require that at least 10% of the species in each super-phylum are present in a group to qualify as bilaterian-specific. They effectively prevent type 3 errors in our list of bilaterian-specific orthogroups that were caused by the addition of <4 sequences. In contrast, we cannot currently prevent potentially wrong orthology inference if four or more sequences of an ancestor-changing lineage were added erroneously (for example, four ecdysozoan sequences added to an otherwise mammalian-specific orthogroup). However, this error mainly affects small bilaterian-

specific orthogroups with only few sequences from deuterostomes, lophotrochozoans, and/or ecdy-sozoans because of their lack in representativeness. Detailed phylogenetic analysis as well as improved taxon sampling would be necessary to discover such false-positive assignments.

Type 4 errors occur if an orthogroup is estimated younger than bilaterians, but is—accidentally—not joined with another, similar orthogroup that would convert the ancestor to Bilateria if combined with the original group (for example, a vertebrate-specific orthogroup and a highly similar insect-specific orthogroup would create a bilaterian-specific orthogroup). To detect such errors, it is necessary to perform all-vs-all profile comparisons of the orthogroups younger than bilaterians. Next, combinations of similar groups need to be determined that would shift the former individual ancestors to a new common bilaterian ancestor and that are each other's bidirectional best hit. Due to the high computational investment we refrained from further investigating this error source in this manuscript.

To further probe accuracy of the 363 bilaterian-specific orthogroups, we mapped human and *D. melanogaster* sequences contained in these orthogroups to the respective genome (versions hg38 and dm6) using BLAT (*Kent, 2002*). Such mapping was possible for 348/363 orthogroups (95.87%). We then checked whether the target gene to which these sequences were assigned, belonged to the initial orthogroup. This was not true in a considerable number of cases. For example, often bilaterian-specific orthogroups contained short ORFs from *H. sapiens* or *D. melanogaster* that mapped to a particular gene. The corresponding full length protein, however, was assigned to a different orthogroup with a different ancestor, indicating that separation of genes into two or more orthogroups affected integrity of the 363 orthogroups set. We therefore excluded all orthogroups with potential mapping inconsistencies and arrived at a set of 204 bilaterian-specific genes. As a final validation step, we blasted at NCBI (non-redundant GenBank version from May 24, 2017) all human or *D. melanogaster* orthologues, which are present in the 204 bilaterian-specific orthogroups, against non-bilaterian metazoans (Metazoa excluding Bilateria and Mesozoa). A reciprocal best hit analysis of the BLAST results indicated that 47 genes, corresponding to 47 orthogroups, might contain orthologues in non-bilaterian species although our orthology prediction pipeline did not detect them. As substantial work is required to confirm or reject these potentially false-positive orthogroups, we removed them from the list and arrived at a final number of 157 orthogroups. These 157 orthogroups represent a minimal set of high-confidence bilaterian-specific orthogroups which is free of most errors present in other orthology databases.

