## [Decision Letter]

**Acceptance summary:**

This paper provides a comprehensive search for proteins whose evolutionary emergence can be dated to the last common ancestor of the Bilateria. The main aim of this study is to shed light on the genetic changes that underlie account of key innovations in the bilaterian body plan. For this purpose, the authors have compiled an impressive data collection comprising protein coding genes from pre-annotated data, a filtered collection of genomic ORFs stemming from a six-frame translation of entire genome sequences, and eventually transcriptome data for taxa lacking a genome sequence. The key asset of this data collection, when compared to existing ortholog databases, is the-according to contemporary standards comprehensive-sampling of non-bilaterian metazoan species. The much more comprehensive inclusion of non-bilaterian datasets in this study allows a more accurate orthogroup clustering than previous studies. Moreover, this database of sequences presented is a rich resource for reconstructing gene content at any metazoan node.

**Decision letter after peer review:**

Thank you for submitting your article "The genetic factors of bilaterian evolution" for consideration by *eLife*. Your article has been reviewed by two peer reviewers, and the evaluation has been overseen by Patricia Wittkopp as the Senior Editor.

The following individuals involved in review of your submission have agreed to reveal their identity: Ingo Ebersberger (Reviewer #2). The reviewers have discussed the reviews with one another and the Senior Editor has drafted this decision to help you prepare a revised submission.

Summary:

In their manuscript, Heger et al. present a comprehensive search for proteins whose evolutionary emergence can be dated to the last common ancestor of the Bilateria. The main aim of this study is to shed light on the genetic changes that underlie account of key innovations in the bilaterian body plan. For this purpose, the authors have compiled an impressive data collection comprising protein coding genes from pre-annotated data, a filtered collection of genomic ORFs stemming from a six-frame translation of entire genome sequences, and eventually transcriptome data for taxa lacking a genome sequence. The key asset of this data collection, when compared to existing ortholog databases, is the-according to contemporary standards comprehensive-sampling of non-bilaterian metazoan species. For the ortholog search, they present a modified version of the orthoMCL algorithm. The resulting orthologous groups are then subjected to a number of intuitive, yet largely ad hoc, filters to identify a set of 157 orthologous groups the authors date to the last common ancestor of the Bilateria. The remainder of the manuscript is then dedicated to discuss these 157 orthologous in the context of bilaterian evolution.

BigWenDB, a new database maximising resolution at the bilaterian origin, building on opisthokont sequences from GenBank, as well as transcriptomes available for non-bilaterian animals, is an important contribution for the field. Orthogroups are identified using a modified the OrthoMCL pipeline, and validated against an external benchmark set of 70 manually curated orthogroups. An HMM-HMM comparison step evaluates orthogroup completeness. The main value of this studies lies in the much more comprehensive inclusion of non-bilaterian datasets in BigWenDB, allowing a more accurate orthogroup clustering than with previous efforts. This database is a rich resource for reconstructing gene content at any metazoan node. Clustering reveals a clear pattern of enrichment among the 157 bilaterian-specific genes, with transcription factors and membrane proteins of various kinds dominating – in contrast to other nodes of the bilaterian tree. Also, the identification of monoamine neurotransmitter reception and nodal signalling as bilaterian-specific innovation is a valuable step forward in our understanding of bilaterian evolution.

Essential revisions:

1) The authors present a way of compiling orthologous groups from their data, which deviates from currently available methods. In particular, the hmm-hmm comparison is, to my knowledge, unprecedented. This pipeline is presented as a main methodological innovation of this manuscript. Although I appreciate that the authors are concerned about the performance of their method, the data set used for method validation is way too small to thoroughly benchmark the method. I strongly suggest that the authors make use of the public benchmark service for ortholog search tools at https://orthology.benchmarkservice.org (Altenhoff et al., 2016) for this purpose.

2) The authors do not consider the limited sensitivity of Blast based ortholog search tools, which leads to an underestimation of gene age (e.g. Elhaik et al., 2006; Luz et al., 2006; Moyers and Zhang, 2015, 2016, 2017; Jain et al., 2019). In this context, it is a bit worrying, that proteins involved in regulatory functions (transcription factors), and proteins located in the membrane appear enriched in the set of Bilateria-specific proteins. Evidence exists for both kinds of proteins that their particular rates and mode of evolution increases the risk of missing distantly related orthologues (Jain et al., 2019). Although the reciprocal hmm-based search could ameliorate this problem, it is likely to be not helpful. This is, because the high evolutionary rate of such proteins paired with the presence of very common functional domains, e.g. a Zn-Finger, probably results in rather uninformative hidden Markov models.

3) I miss the definition of clear-cut evolutionary scenarios. For example, it is unclear what it means when a gene emerges in the LCA of the Bilateria. Is it a de-novo gene birth? Or do the authors also take into account lineage specific gene duplications, probably in combination with domain gain or loss? From what I see in the data I have the impression that the authors consider both. I will exemplify my concern using Figure S14. It shows a phylogeny comprising both FoxH1-an alleged bilaterian invention-and the older FoxD. The way how the tree is drawn implies that it is rooted. If so, it can only be interpreted in the following way: A gene duplication at the root of the metazoa gave rise to FoxH1 and to FoxD. While FoxD is still seen in metazoa, which branched off prior to bilaterian diversification, FoxH1 is nowadays retained only in the Bilateria, and the corresponding orthologs were lost in earlier branching taxa. If this were true, then FoxD would be as old as the animals. Alternatively, one could consider an outgroup rooting. Then FoxD and FoxH1 diversified only at the root of the Bilateria. This diversification would then be the true innovation, but then FoxD and FoxH1 would be co-orthologous to the proteins in the early branching animals. I think the authors have to be way more specific here.

4) The authors use phylogenetic trees to support, or more often to reject an orthology relationship, e.g. subsection “Changes in axon guidance accompany bilaterian evolution”. I am missing topology tests, e.g. the Shimodaira Hasegawa test, to show that a gene tree indeed differs significantly from the species tree. This is necessary, because the phylogenetic information within one orthologous group is, in many cases, not sufficient for accurately resolving the evolutionary history of the corresponding proteins.

5) The resampling approach to show that bilaterian genes are significantly more connected in PPI networks than it is expected by chance is likely to be confounded with gene age. There is evidence that evolutionarily older genes are more central in PPI networks than younger ones (Kim and Marcotte, 2008). Any resampling method that also selects young genes is therefore likely to result in less connected networks.

6) The reciprocal best hit approach, which is used in the final filtering of the candidates is problematic. Reason is, that the initial Blast against non-redundant genbank can have any paralog as a best hit, since the assumption that the entire proteome of each species is represented in the data base is probably not valid.

7) Based on what evidence do the authors consider five independent losses unlikely? And what is an “undetermined orthology”. Can the authors reject that the poriferan sequence is an ortholog of Eomes, and if so again based on what evidence?

8) To consider: one reviewer found the data set overly complex. Because the story is pretty complex, they thought it would do very little harm to omit the genomic ORFs from the analysis.

[Editors' note: further revisions were suggested prior to acceptance, as described below.]

Thank you for resubmitting your work entitled "The genetic factors of bilaterian evolution" for further consideration by *eLife*. Your revised article has been evaluated by Patricia Wittkopp (Senior Editor), and two reviewers, including the guest Reviewing Editor.

The manuscript has been improved but there are some remaining issues that need to be addressed before acceptance, as outlined in the comments from the re-review below.

Reviewer #1:

This is an interesting and important manuscript dealing with the identification and reconstruction of the ancestral pool of bilaterian genes and bilaterian innovations.

Different authors used different approaches, and the story is ongoing. I consider the manuscript as a next step to decipher the complex story of the amazing diversification among bilaterians.

In my opinion, it is also a valuable reference paper, and this study would be used by the biological research community for future efforts to reconstruct the set of genes underlying bilaterians innovations.

The paper is supported by an extensive supplement, which is an additional and beneficial reference source of specific information about hundreds of genes.

The authors performed a very careful revision of the manuscript and clarified the previously raised questions. The authors identified a subset of about 150 genes as bilaterian-specific innovations and provided a solid argument to support their gene selection criteria.

In general, the paper advances the field and will be an important web resource of comparative data for future investigations.

One of the major critical points: The authors mainly provide the description of identified genes and relatively little critical analyses. I understand that the selected design of the manuscript might reflect the nature of the complex bioinformatic study and reflect a limited knowledge about the functions of many genes in different bilaterian lineages. Nevertheless, I would recommend to critically reread the manuscript and carefully think about generalized statements, when they use references to the vertebrate/arthropod data vs. other lineages.

There are some moderate and minor comments, which might improve this manuscript.

1) Introduction: "In contrast, the evolutionary relationships of non-bilaterian metazoans are still a matter of debate, in particular the relative positions of placozoans, ctenophores, and sponges (Brooke and Holland, 2003; Ryan et al., 2013; Pisani et al., 2015; Simion et al., 2017; Feuda et al., 2017). "

– It is a biased reference list with three references stressing one point; the alternative reconstruction of the animal phylogeny should also be presented (e.g., Whelan et al., 2015 – PNAS; Whelan et al., 2017)

2) Introduction: Genomic sequencing of non-bilaterian animals revealed that essentially all signalling pathways and developmentally important genes from bilaterians are also present in non-bilaterians, indicating that these genes evolved before the advent of bilaterians.….

– "…all…important genes" is an incorrect and biased statement. Frankly, we do not know all important genes for most of the bilaterian lineages. Regarding all signaling pathways, many genes related to neuronal signaling and immune systems, for example, are absent in some non-bilaterian metazoans (see for examples in Moroz and Kohn, 2016)

3) Result subsection “Bilaterian G protein-coupled receptors and the control of physiological state through circulatory flow”, paragraphs one and two.

– Discussion about the monoaminergic reward system, as a generalized statement for bilaterians, is highly speculative. I would recommend using more careful wordings since the functions of many GPCRs are putative and speculative.

-

4) Result subsection “Bilaterian G protein-coupled receptors and the control of physiological state through circulatory flow”, paragraph three .

– Speculations about the peptidergic reception. There are many uncertainties about GPCRs in the majority of bilaterians, and it is difficult to say about functions and ligands of putative (neuro)peptide receptors. I suggest using more careful wording.

Reviewer #2:

In their revised version, the authors have addressed many of my initial concerns, and I trust that this is a relevant study, in principle.

However, some issues remain, which I think can be addressed by carefully re-working the argumentation.

1) I asked in my initial review about a benchmark of the new ortholog assignment method making use of the QfO reference proteome set. The authors chose to not follow this up and justified this with two arguments.

a) First, they stated that their core method is basically a re-implementation of OrthoMCL, and use the times the OrthoMCL paper has been cited as kind of a quality criterion. Needless to say that I am not happy with this argument. OrthoMCL has been shown to have, compared to other methods, a very high false positive rate of 16% (Chen et al., 2007). Thus, it is probably not the number of citations for OrthoMCL, but rather the performance of this tool, the authors should use in their argumentation. I strongly encourage to adapt the discussion in this direction. If they have the feeling that a high false positive rate does not matter, which of course is the case when a high sensitivity is desired, and false positives are either sorted out at a later curation step, or false positive make the analysis more conservative, and thus are ok, then they should explain this to the reader.

b) As a second argument, the authors state that the key advantage „(...) is not the core method (OrthoMCL), but the inclusion of several steps for error correction after orthology clustering, including a new HMM-HMM reciprocal best hit test for orthogroup completion (see, e.g., Supplementary file 1—supplementary table 7). These additional steps (together with careful sampling) distinguish our pipeline from the original OrthoMCL and from other orthology pipelines and influence the interpretation of cluster/orthogroup origin. The additional steps are computationally demanding and require, at present, extensive manual processing. We performed them, as proof-of-principle, for the 157 orthogroups with assumed bilaterian ancestry, but this approach is not yet scalable. Applying the steps for error correction to all clustered outputs for reference proteomes from a public database is therefore not feasible."

I am a bit puzzled reading this response. If I understand the answer correctly then the authors state that they perform a semi-automated curation of orthologous groups predicted by orthoMCL, which is absolutely fair. In their manuscript, however, the authors advertise their method as "A unique orthology pipeline for the identification of bilaterian-specific genes", which creates the (untested) impression that the pipeline outperforms existing ortholog search tools. I see two possibilities. Either, the authors present their method as a curation step without the claim of having developed a new ortholog search pipeline. Or, alternatively, they perform a comprehensive benchmark comparing it to existing and state of the art “orthology pipelines”. As a last aspect to consider, part of the QfO benchmark analyses are independent of the QfO reference proteomes. For example, tree discordance tests can be, in principle performed with any underlying set of proteomes.

To sum it up, my main concern is not necessarily the method itself, except the orthologous group completion step using hmm-hmm searches, of course. Rather, I am very much worried about the fact that the authors generate the impression of having developed a new pipeline for orthologous group compilation, which apparently performs better than existing software.

2) My second concern was the limited sensitivity of Blast in ortholog searches. I very much appreciate that the authors have considered this now in the Discussion. In my opinion, some aspects require further attention. Moyers and Zhang, as well as Jain et al. have shown that the sensitivity of Blast can become limiting even when searching for orthologs in the same kingdom. I think it would be a good idea to openly discuss the issue of limited sensitivity ignoring the fact that you “only” search within the same kingdom. It is then straightforward to propose extensive taxon sampling together with the very sensitive hmm-hmm comparisons as a way out of this problem. The shift to a different search method comes then at the cost of a substantially decreased specificity in the ortholog assignment (see my point above). The authors can then explain why they think that an elevated rate of false positive either do not harm, or how they get rid of them in a downstream curation.

3) The authors write in their response "Our primary intention was to generate molecular phylogenies as a means of validating orthogroup clustering, not for inferring specific evolutionary scenarios directly from the gene trees." I cannot follow this argumentation, and I am convinced that I misinterpret the answer. Phylogenetic trees are the basis for deriving evolutionary scenarios on the sequence level, and I cannot see how this cannot be the case. In this context, I regret to say that I find the sentence "The relationship between orthology clusters is imposed by the method and does not necessarily reflect the timing of evolutionary events" impossible to understand. A phylogenetic tree that does not reflect the timing of evolutionary events is either wrong or unrooted. If the latter is the case, then there has to exist the possibility to place the root such that the evolutionary scenario emerges that is proposed in the text. If this is not the case, then again either the tree or the proposed scenario is wrong.

Concerning the precise example of FoxH1: Given the tree shown in Figure 5—figure supplement 3, there is no way that FoxH1 is a bilaterian invention. The FoxH1 lineage split from its sister group (either FoxQ1 or FoxD) via a gene duplication prior to the diversification of the bilateria. If no FoxH1 sequences outside the Bilateria were found, then they were either lost-which still does not make FoxH1 a bilaterian invention-or the taxon sampling was not sufficient, and non-Bilaterian FoxH1 sequences await their detection. Alternatively, the tree is wrong, but then it should not be shown. I consider this a major issue.

4) As a response to my concern about missing topology tests, the authors have recomputed the trees and have added approximate SH-LRT support values. They write in their answer "Although the new phylogenies fully support the conclusions derived from the original analysis using RAxML, we acknowledge that some trees show high gene/species topology discordance within a given protein clade(orthogroup). We accept that this is, in part, the consequence of including mixed data types, such as the shorter ORF sequences, and sequence sets not optimized for phylogeny because they are often derived directly from the orthogroups obtained in our clustering." I do not get the point about why it is a problem of using sequences directly derived from orthogroups. Is this again an issue of missing specificity in the ortholog assignments?

[Editors' note: further revisions were suggested prior to acceptance, as described below.]

Thank you for submitting your article "The genetic factors of bilaterian evolution" for consideration by *eLife*. Your article has been considered again by one of the original reviewers, and the evaluation has been overseen by Patricia Wittkopp as the Senior Editor. While we found most of the reviewers’ prior concerns addressed satisfactorily, there was also one major concern not fully addressed. This concern must be addressed in an additional revision before we can consider this manuscript further.

This is a highly complex manuscript, and all conclusions essentially depend on the accuracy of the orthology assignment and the correct phylogenetic interpretation. This was pointed out in the previous round of reviews using the FoxH1 tree as an example. This tree suggests that FoxH1 is as old as the metazoa, and the same is true for EOMES (Figure 5—figure supplement 3 and 4). In the response, the authors state that there are several reasons why the trees shown might not accurately reflect the evolutionary history, and mainly mention heterotachy following gene duplication, which makes the gene look older than it is. This may or may not be correct. To further corroborate their argumentation, the authors state that independent gene loss, which would be required to explain the present day distribution of the respective genes is unlikely, and there is no report for such parallel losses in the literature. Previously published work, including https://www.biorxiv.org/content/10.1101/2020.04.23.058008v1 and https://www.nature.com/articles/s41467-018-03667-1 are at odds with this statement.

We recognize that reconstructing evolutionary relationships with individual sequences is hard. But if the tree is at odds with the conclusion, then this has to be addressed with analyses, and not just by saying “probably the tree is wrong”.

We see two ways out of this dilemma:

One is for you to perform additional analyses, which will take time. This would likely have to include the results of the ortholog searches in a well-defined set of species, such that presence/absence of orthologs can be interpreted along a (known species) tree. For example, at the moment, it is unclear why the tree of Nodal features about 30 vertebrate sequences, that of FoxH1 only about 20 sequences, and that of EOMES only 2. Although these numbers are not relevant when it comes to the age of the genes, they tell us something about the precision and rigor of the analysis.

The other possibility is that the authors could modify the manuscript to avoid the term “innovation” where it is not backed up by the data. If the tree argues for an evolutionarily old gene, which nowadays – and according to the analyses of the authors – is confined to the bilateria, why not simply call it a “bilaterian-specific” gene, leaving it for the moment open whether it was indeed an “innovation” or a selective retention of an older gene. In essence, the authors would then argue with their phylogenetic profiles, which seem to be comprehensive, and not with trees, which they indicate in the response might be wrong anyways. If the authors then wish, they could speculate that the gene is indeed a bilateral innovation, but it has to be clear that this is a speculation that would require more thorough analyses to be tested.

---

## [Author Response]

Essential revisions:1) The authors present a way of compiling orthologous groups from their data, which deviates from currently available methods. In particular, the hmm-hmm comparison is, to my knowledge, unprecedented. This pipeline is presented as a main methodological innovation of this manuscript. Although I appreciate that the authors are concerned about the performance of their method, the data set used for method validation is way too small to thoroughly benchmark the method. I strongly suggest that the authors make use of the public benchmark service for ortholog search tools at https://orthology.benchmarkservice.org (Altenhoff et al., 2016) for this purpose.

The reviewers address an important question concerning the reliability of our orthology groupings and suggest that we benchmark our method using a public benchmark service for orthology search tools (https://orthology.benchmarkservice.org). This service provides reference proteomes of 78 archaea, bacteria, and eukaryotes (~1.3 million sequences) for processing in newly developed orthology pipelines, and the resulting ortholog clusters are uploaded to the benchmark service website for comparison with other pipelines and statistical analyses of method accuracy. We have used the reviewers’ suggestion as an opportunity to explore several aspects of dataset and method validation. However, the result is that we refrained from using this particular service for several reasons:

First, our core method for orthology inference is identical to OrthoMCL (Li et al., 2003), a well-established and widely used method (>3700 citations in Google Scholar). In our work the only difference to the original OrthoMCL method is porting the code (Perl and MySQL) to the statistical programming language R to be able to process a larger number of sequences. We show in the manuscript that the original and our Rbased pipeline produce identical results (Supplementary file 1—supplementary table 4) and therefore we cannot expect clustering differences between the two versions.

Importantly, the key advantage of our workflow is not the core method (OrthoMCL), but the inclusion of several steps for error correction after orthology clustering, including a new HMM-HMM reciprocal best hit test for orthogroup completion (see, e.g., Supplementary file 1—supplementary table 7). These additional steps (together with careful sampling) distinguish our pipeline from the original OrthoMCL and from other orthology pipelines and influence the interpretation of cluster/orthogroup origin.

The additional steps are computationally demanding and require, at present, extensive manual processing. We performed them, as proof-of-principle, for the 157 orthogroups with assumed bilaterian ancestry, but this approach is not yet scalable. Applying the steps for error correction to all clustered outputs for reference proteomes from a public database is therefore not feasible. We would be happy to help develop automatic solutions for these steps in the future to make the method accessible for a wider community.

Moreover, any clustering pipeline heavily depends on the input dataset, and as the reviewers note a key innovation is the content of our BigWenDB. What we really want, to increase confidence in our data, is to validate, by using other methods, the clustering results OrthoMCL produced with OUR dataset. To accomplish this, we would have to process our dataset in parallel with other orthology pipelines and compare the results. This, however, is a major project that is beyond the scope of the current study, as evidenced by calls for dedicated benchmarking studies (e.g., https://www.biomedcentral.com/collections/benchmarkingstudies) and joint efforts to benchmark, improve and standardize orthology predictions (https://questfororthologs.org/).

Nevertheless, we agree that additional validation is desirable and searched for public datasets that may also contain bilaterian-specific gene predictions. We first considered the well-established orthology database OrthoDB version 10 (orthodb.org; Kriventseva et al., 2018). Although this database provides orthogroup analyses for different taxonomic levels (nodes) of the phylogenetic tree, in fact each independent clustering analysis is based on input data of the given taxon only, whereas the identification of taxon-specific orthogroups requires simultaneous analysis of all taxa followed by assessment of taxon membership per orthogroup. It is therefore not possible to obtain lineage-specific orthogroup sets from this database (e.g. metazoan-specific genes) or from BUSCO sets (lineage-specific sets of Benchmarking Universal Single-Copy Orthologs; Waterhouse et al., MBE, 2017) derived from OrthoDB.

The same is true for TreeFam (http://www.treefam.org) which is used by the above mentioned orthology benchmark service. While a fraction of our bilaterian-specific genes also appear as bilaterian-specific in TreeFam (20 of 50 tested), the power of TreeFam to distinguish bilaterian from eumetazoan or metazoan origin is low, as it contains only a single cnidarian outgroup species (BigWenDB: 19 species) and two other non-bilaterians (BigWenDB: 14), and it might have difficulties to distinguish vertebrate/chordate-specific orthogroups from bilaterian-specific groups due to the shortage in lophotrochozoan species (3; BigWenDB: 18). While we do not question the quality of the TreeFam data, they are not ideal for validating bilaterianspecific or metazoan-specific genes.

We then sought to compare our data with a dataset compiled by Paps and Holland, 2018. In this paper, the authors reconstructed the ancestral metazoan genome by orthology clustering of 62 metazoan and outgroup genomes, conceptually similar to our study. They provide in their Supplementary Material lineage-specific genes not only for the metazoan node, but also for the bilaterian and other nodes (https://www.nature.com/articles/s41467-018-04136-5#Sec17; file: 41467_2018_4136_MOESM13_ESM.zip). There are 1,580 bilaterian-specific orthogroups in the file "Bilateria Novel all genes IDs.out" of the Paps and Holland dataset. As many of the orthogroups are small and contained only a few species, we filtered these away and retained OGs with better coverage (at least 15 different species), including a human ortholog. Of the 92 OGs fulfilling these criteria, 15 had overlap with our 157 bilaterian-specific genes. However, as this study did not consider orthogroup composition (phylogenetic coverage) and did not carry out error correction steps, their results cannot be taken as an established reference for the validation of our data.

In summary, we think that the suggested benchmark test does not provide the desired validation of our approach. Rather, with the BigWenDB and our analysis approach we offer a rich new resource, including our orthology clustering predictions, as a baseline for future testing of alternative methods. To support this, we deposited our results and sequence dataset at datadryad.org (temporary review link: https://datadryad.org/review?doi=doi:10.5061/dryad.4qf7168) and the R code for OrthoMCL at https://github.com/BigWenDB (still private, will be made public after acceptance of the paper). We also note that we have done due diligence in comparing our results against 70 curated orthogroups and in conducting a number of phylogenetic tests with molecular outgroups to validate our clustering results (see also response #3-4 below).

2) The authors do not consider the limited sensitivity of Blast based ortholog search tools, which leads to an underestimation of gene age (e.g. Elhaik et al., 2006; Luz et al., 2006; Moyers and Zhang, 2015, 2016, 2017; Jain et al., 2019). In this context, it is a bit worrying, that proteins involved in regulatory functions (transcription factors), and proteins located in the membrane appear enriched in the set of Bilateria-specific proteins. Evidence exists for both kinds of proteins that their particular rates and mode of evolution increases the risk of missing distantly related orthologs (Jain et al., 2019). Although the reciprocal hmm-based search could ameliorate this problem, it is likely to be not helpful. This is, because the high evolutionary rate of such proteins paired with the presence of very common functional domains, e.g. a Zn-Finger, probably results in rather uninformative hidden Markov models.

The articles mentioned by the reviewers show that Blast-based orthology search tools tend to underestimate gene age if the most distant (earliest branching) true homolog is missed because of Blast's limited sensitivity or because of a gene's fast evolution. The reviewers specifically mention the study of Jain et al., 2019, in which the authors investigated the concept of protein traceability, i.e. the "evolutionary distance beyond which sequences are too diverged to be detected with BlastP-based ortholog search tools". Using yeast genes and ortholog searches across different kingdoms (Bacteria, Archaea, protists, plants, green algae, Metazoa), the authors show that protein traceability is high when performing intra-kingdom comparisons (their Figure 2) and considerably lower when performing inter-kingdom comparisons. As the focus of our study is the border between bilaterian and non-bilaterian metazoans, the critical comparisons take place within the same kingdom (Metazoa), and issues caused by low protein traceability should therefore be less prominent than in the scenarios outlined by Jain et al., 2019.

In addition, Jain et al. demonstrate that protein traceability changes with the underlying training data: with phylogenetically diverse training data, better recovery of distant homologs is possible. Again, this principle is met in our study as we used ortholog alignments across the entire bilaterian lineage for the recovery of distant homologs in HMM-HMM comparison steps. With the inclusion of many different bilaterian and nonbilaterian species and the evaluation of orthogroup composition in a phylogenetic context, our dataset further minimizes errors in inferring orthogroup origin that may be caused by the low traceability of specific genes or by taxa with particularly high evolutionary rates. We added a new paragraph discussing this.

As for the reviewers’ concerns for specific gene classes, comparative studies in non-bilaterian metazoans revealed that much of the developmental signaling and transcription factor repertoire was already present in the metazoan stem (see, e.g., Lanna, Evo-devo of non-bilaterian animals, Genet Mol Biol, 2015, and references therein). The successful identification of these factors in non-bilaterian metazoans suggests that traceability within Metazoa does not seem to be a general problem of a metazoan-centered dataset like ours. In the revised version, we briefly discuss these issues in the Discussion of the manuscript.

Nevertheless, it is possible that in particular cases low traceability causes wrong inferences, and the reviewers acknowledged that our newly developed reciprocal best-hit HMM-HMM search step could improve orthogroup assignments in such cases. However, the reviewers also expressed concern that the "high evolutionary rate of such proteins paired with the presence of very common functional domains, e.g. a Zn-Finger, probably results in rather uninformative hidden Markov models."

To address this, we have now generated sequence logo visualisations of two example orthogroups with a common functional domain, the poly-zinc finger domain (new figure: Supplementary file 1—supplementary figure 2) and mention these in the text. While it is true that these HMMs (or sequence logo visualisations) are uninformative in areas outside the poly-ZF region, their central conserved poly-ZF domain is highly informative. The sequence logos not only demonstrate the high conservation of Cys and His residues (Zn^2+^ complexing) common to both proteins, but also show that individual Zinc fingers of each protein have highly conserved and unique profiles distinguishing them from other ZF proteins. We therefore think that the HMMs generated from such alignments are informative in the sense that they capture the essence (the important positions and their spacing and variability) of an alignment/orthogroup. Indeed, we demonstrated in a number of examples that the approach is capable of linking previously separate (and sometimes taxonomically complementary) orthogroups on the basis of their HMM profiles (Supplementary file 1—supplementary tables 7, 14, 16, 18, 24). For example, lophotrochozoan orthologs of FoxH1 fall into two orthogroups of FoxH1 proteins (one with mammals and the other with fishes and ecdysozoans), such that our HMM-HMM analysis and taxonomic breadth accurately recovers and unifies these orthogroups for a bilaterian-wide protein.

The HMM-HMM comparison step also allowed us to identify potential orthologs of bilaterian-specific proteins in non-bilaterian species, leading to the exclusion of these candidates and demonstrating the rigor of this approach for determining meaningful sequence similarity.

3) I miss the definition of clear-cut evolutionary scenarios. For example, it is unclear what it means when a gene emerges in the LCA of the Bilateria. Is it a de-novo gene birth? Or do the authors also take into account lineage specific gene duplications, probably in combination with domain gain or loss? From what I see in the data I have the impression that the authors consider both. I will exemplify my concern using Figure S14. It shows a phylogeny comprising both FoxH1-an alleged bilaterian invention-and the older FoxD. The way how the tree is drawn implies that it is rooted. If so, it can only be interpreted in the following way: A gene duplication at the root of the metazoa gave rise to FoxH1 and to FoxD. While FoxD is still seen in metazoa, which branched off prior to bilaterian diversification, FoxH1 is nowadays retained only in the Bilateria, and the corresponding orthologs were lost in earlier branching taxa. If this were true, then FoxD would be as old as the animals. Alternatively, one could consider an outgroup rooting. Then FoxD and FoxH1 diversified only at the root of the Bilateria. This diversification would then be the true innovation, but then FoxD and FoxH1 would be co-orthologous to the proteins in the early branching animals. I think the authors have to be way more specific here.

We apologize to the reviewers for not explicitly stating our assumptions and interpretations of the phylogenetic trees (and see also our response to the next point #4, below). Our primary intention was to generate molecular phylogenies as a means of validating orthogroup clustering, not for inferring specific evolutionary scenarios directly from the gene trees. The Fox gene phylogeny example highlighted by the reviewer (original Figure S14) includes all members of the two orthogroups that contain putative FoxH1 proteins, and we included a taxonomically broad subset of proteins from the nearest molecular outgroup in our dataset (the orthogroup with FoxD proteins). The phylogenetic tree recovers 100% support for FoxH1 and FoxD proteins as forming protein-specific, distinct clades. Thus, we are confident in identifying all members of the two FoxH1 orthogroups as genuine FoxH1 orthologs. Furthermore, in re-evaluating this particular example in order to reply to the reviewers, we revisited the primary literature on Fox genes, which – with smallerscale phylogenetic sampling than in our dataset – implicated FoxQ1 proteins as being the nearest outgroup to FoxH1. We have thus repeated our phylogenetic analyses with both FoxD and FoxQ1 outgroups, and again strongly recover a single FoxH1 clade (new Figure 5—figure supplement 3). The evolutionary scenario of FoxH1 being ascribed to the bilaterian ancestor then derives directly from the taxonomic membership within the FoxH1 orthogroups.

To avoid potential misunderstandings, we now modified the legends of phylogenetic trees throughout the manuscript by adding the sentence: "The relationship between orthology clusters is imposed by the method and does not necessarily reflect the timing of evolutionary events".

As for the precise nature of evolutionary novelty, the reviewer is correct that the 157 bilaterian-specific proteins span genuinely de novo proteins as well as – predominantly – those that originated by major evolutionary changes such as gene duplication and protein domain changes. In fact, we only identify 5 of the 157 bilaterian-specific proteins as being likely de novo gene births and mention this now in the text (supported by alignments in Figure 2—figure supplement 2, 3, 4). HMMs corresponding to these 5 orthogroups did not have hits outside Bilateria (Supplementary file 1—supplementary table 9). In contrast, the majority of bilaterian-specific proteins contained one or more known protein domains.

4) The authors use phylogenetic trees to support, or more often to reject an orthology relationship, e.g. subsection “Changes in axon guidance accompany bilaterian evolution”. I am missing topology tests, e.g. the Shimodaira Hasegawa test, to show that a gene tree indeed differs significantly from the species tree. This is necessary, because the phylogenetic information within one orthologous group is, in many cases, not sufficient for accurately resolving the evolutionary history of the corresponding proteins.

Following on from point #3 above, we agree that our original phylogenies were not optimized for directly deducing evolutionary scenarios. Instead, we use phylogenies for testing the recovery of orthogroups as well-supported molecular clades and for investigating whether or not outlier sequences from phylogenetically distant species were erroneously included into an orthogroup. For these purposes, formal topology tests for gene-tree species-tree concordance within orthogroups are, in our opinion, not strictly necessary.

With respect to the reviewers' concern, we recalculated all phylogenetic trees of this article with a new method (IQ-Tree) and included Shimodaira–Hasegawa-like approximate likelihood ratio tests (SH-aLRT) whose results are now displayed on all branch labels of the trees, in addition to bootstrap values. According to the IQ-Tree documentation, one would typically rely on a clade if its SH-aLRT value is > = 80% and its UFboot value is > = 95%, conditions that are met for the relevant nodes in all our analyses.

Although the new phylogenies fully support the conclusions derived from the original analysis using RAxML, we acknowledge that some trees show high gene/species topology discordance within a given protein clade (orthogroup). We accept that this is, in part, the consequence of including mixed data types, such as the shorter ORF sequences, and sequence sets not optimized for phylogeny because they are often derived directly from the orthogroups obtained in our clustering.

5) The resampling approach to show that bilaterian genes are significantly more connected in PPI networks than it is expected by chance is likely to be confounded with gene age. There is evidence that evolutionarily older genes are more central in PPI networks than younger ones (Kim and Marcotte, 2008). Any resampling method that also selects young genes is therefore likely to result in less connected networks.

We thank the reviewers for identifying this important point. To investigate whether the high connectedness of bilaterian-specific genes is indeed biased by age, we carried out additional analyses.

First, we obtained from our clustering results a list of 5,429 (of 20,205) human genes/orthogroups with broad phylogenetic distribution (present in >75 species and unique mapping) and reliable birth in the ancestor of bilaterians or older. We then selected 50x 150 out of the 5,429 genes and analysed their PPI network at stringDB. The obtained PPI score distribution showed that the value for bilaterian-specific genes fell into a broad peak of PPI scores of these older genes and was thus not indicative of a particularly high connectedness in bilaterian genes.

In this first approach, most of the 5,428 considered genes/orthogroups originated in the ancestor of opisthokonts, the oldest possible phylostratum in our database. To avoid this bias, we selected from the corresponding list a set of 797 genes/orthogroups with metazoan ancestor and compared this age-stratified gene set with the bilaterian-specific gene set. We selected 100x 150 genes/orthogroups from the 797 metazoan-specific gene set and analysed their PPI score at stringDB. This time, the PPI score of our bilaterianspecific gene set was much lower than the peak of the metazoan-specific PPI distribution, indicating a clear difference in connectedness. However, we were not satisfied with this result because in this case we compared a "full" gene set (bilaterian-specific OGs) with a distribution of subsets (100x 150 out of 797), in which existing PPI networks might have been destroyed by subsampling.

We then analysed in detail the interactions of bilaterian-specific and metazoan-specific genes in a combined network of the two sets and found support for a higher level of connectivity for bilaterian-specific proteins, in comparison to metazoan-specific proteins. These analyses are described in a new paragraph on PPI networks and are accompanied by a new Figure 8 (previous Figure 6) and new corresponding method paragraphs.

The above described results were obtained with stringDB version 11, which is currently running on the stringDB webserver. As our original network graph was obtained with an earlier version of stringDB (version 10.5) and a slightly incomplete set of bilaterian-specific genes (we missed to add OSR1, NTRK1, EOMES), we re-analysed interactions using the new stringDB version and a complete gene set.

This reanalysis, however, led to the new conclusion that the previously proposed hub gene PRDM10 was a false positive result. In the new database version, all interactions of PDRM10 with other proteins were removed, even for experimentally validated interactions. We contacted the stringDB support team with this issue and they explained that "textmining links in version 10 for PRDM10 are false positives due to PRDM10 synonym Tris, which is also part of the name of the buffer. In version 11 we, most likely, blacklisted that synonym".

We therefore removed the paragraph on PDRM10, including the wet bench expression, RNAi data, the PPI control figure (original Figure S21), and corresponding methods, from the manuscript, as it is no longer supported by stringDB interaction data.

6) The reciprocal best hit approach, which is used in the final filtering of the candidates is problematic. Reason is, that the initial Blast against non-redundant genbank can have any paralog as a best hit, since the assumption that the entire proteome of each species is represented in the data base is probably not valid.

As final validation step for the bilaterian-specific genes, we blasted all human or *D. melanogaster* orthologs, which were present in a precursor list of 204 bilaterian-specific orthogroups, against non-bilaterian metazoans at NCBI. If a reciprocal best-hit analysis was positive, it suggested that potential orthologs of the bilaterian-specific gene query might exist in non-bilaterians. To avoid the risk of including false positives, we therefore decided to remove such candidates from the set of bilaterian genes, irrespective of their true relationship. We accepted these losses because final proof would require more reliable sequence data from non-bilaterians and detailed additional analyses, or wet lab experiments.

7) Based on what evidence do the authors consider five independent losses unlikely? And what is an “undetermined orthology”. Can the authors reject that the poriferan sequence is an ortholog of Eomes, and if so again based on what evidence?

To clarify our arguments for a bilaterian origin of the Nodal pathway target Eomes, we rewrote the corresponding paragraph and included as further support for our initial conclusion an additional table with reciprocal best hit Blast analysis (new Supplementary file 1—supplementary table 15) of the two putative poriferan Eomes sequences.

8) To consider: one reviewer found the data set overly complex. Because the story is pretty complex, they thought it would do very little harm to omit the genomic ORFs from the analysis.

We explained in the manuscript why we have chosen to include a metazoan-wide set of genomic ORFs (missing, unreliable and/or erroneous genome annotation, especially in non-model organisms). The decision to include ORFs led to the current complexity of the dataset. All analyses are based on the ORF-containing dataset, and we do not see a productive way to remove the ORFs without having to rerun all analyses.

Instead, we think that the addition of ORFs is valuable. For example, the bilaterian origin of FoxH1, and consequently of the entire Nodal signalling pathway, would not have been detected without the inclusion of genomic ORFs. ORFs represent a substantial fraction of sequences in many orthogroups, and they contributed significantly to the generation of ortholog clusters during the clustering steps.

Furthermore, the genomic ORFs gave rise to ORF-only orthogroups (not mentioned in the manuscript), with the possibility to detect new lineage-specific protein-coding factors that so far escaped attention. The analysis of these orthogroups is currently under way.

[Editors' note: further revisions were suggested prior to acceptance, as described below.]

Reviewer #1:[…] There are some moderate and minor comments, which might improve this manuscript.1) Introduction: "In contrast, the evolutionary relationships of non-bilaterian metazoans are still a matter of debate, in particular the relative positions of placozoans, ctenophores, and sponges (Brooke and Holland, 2003; Ryan et al., 2013; Pisani et al., 2015; Simion et al., 2017; Feuda et al., 2017). "– It is a biased reference list with three references stressing one point; the alternative reconstruction of the animal phylogeny should also be presented (e.g., Whelan et al., 2015 – PNAS; Whelan et al., 2017)

Thank you for your comment. As suggested, we now include Whelan et al., 2017.

2) Introduction: Genomic sequencing of non-bilaterian animals revealed that essentially all signaling pathways and developmentally important genes from bilaterians are also present in non-bilaterians, indicating that these genes evolved before the advent of bilaterians.….– "…all…important genes" is an incorrect and biased statement. Frankly, we do not know all important genes for most of the bilaterian lineages. Regarding all signaling pathways, many genes related to neuronal signaling and immune systems, for example, are absent in some non-bilaterian metazoans (see for examples in Moroz and Kohn, 2016)

We agree that our original sentence contained an inappropriate generalization. The seven major signalling pathways present in metazoans are: TGF-β, canonical Wnt, nuclear receptors, Notch-Δ, Hedgehog, RTK/growth factors, JAK/STAT, and extracellular matrix, and essentially all of these exist also in non-bilaterians (see Babonis and Martindale, Phylogenetic evidence for the modular evolution of metazoan signalling pathways, 2016, Phil. Trans. R. Soc. B 372: 20150477. http://dx.doi.org/10.1098/rstb.2015.0477). To avoid misunderstandings, we changed "essentially all" to "the major" and inserted "many" before "developmentally important genes from bilaterians".

3) Result subsection “Bilaterian G protein-coupled receptors and the control of physiological state through circulatory flow”, paragraphs one and two.– Discussion about the monoaminergic reward system, as a generalized statement for bilaterians, is highly speculative. I would recommend using more careful wordings since the functions of many GPCRs are putative and speculative.

Indeed, our initial references (Berridge, 2004; Berger et al.,2009) only referred to the function of monaminergic signaling in humans/mammals. However, there is a large body of literature supporting the view that dopamine, adrenaline, and serotonin signalling indeed regulate many important processes (sleep, arousal, wakefulness, aggression, locomotion, feeding and courtship behaviour, memory formation and learning) in several bilaterian phyla, e.g. in arthropods, nematodes, mollusks, and vertebrates. Although these neurotransmitters have not been studied in all bilaterian phyla, they are involved in similar physiological processes in a range of different phyla. To take this into account, we modified the text and included additional references from non-vertebrate model systems.

We agree that also reference Budhiraja, 2009, in the following paragraph, does not fully document what we wanted to express as this article only refers to the Neuromedin-U receptor in humans. However, we surveyed the literature again (Feb 2020) and we are still confident that our initial statements were justified as we found additional supporting examples in various bilaterian phyla. We reworded the paragraph and included several new references to monoamine signalling across bilaterians.

4) Result subsection “Bilaterian G protein-coupled receptors and the control of physiological state through circulatory flow”, paragraph three.– Speculations about the peptidergic reception. There are many uncertainties about GPCRs in the majority of bilaterians, and it is difficult to say about functions and ligands of putative (neuro)peptide receptors. I suggest using more careful wording.

We re-edited this paragraph and incorporated references that better support our initial statements.

Reviewer #2:In their revised version, the authors have addressed many of my initial concerns, and I trust that this is a relevant study, in principle.However, some issues remain, which I think can be addressed by carefully re-working the argumentation.1) I asked in my initial review about a benchmark of the new ortholog assignment method making use of the QfO reference proteome set. The authors chose to not follow this up and justified this with two arguments.a) First, they stated that their core method is basically a re-implementation of OrthoMCL, and use the times the OrthoMCL paper has been cited as kind of a quality criterion. Needless to say that I am not happy with this argument. OrthoMCL has been shown to have, compared to other methods, a very high false positive rate of 16% (Chen et al., 2007). Thus, it is probably not the number of citations for OrthoMCL, but rather the performance of this tool, the authors should use in their argumentation. I strongly encourage to adapt the discussion in this direction. If they have the feeling that a high false positive rate does not matter, which of course is the case when a high sensitivity is desired, and false positives are either sorted out at a later curation step, or false positive make the analysis more conservative, and thus are ok, then they should explain this to the reader.b) As a second argument, the authors state that the key advantage „(...) is not the core method (OrthoMCL), but the inclusion of several steps for error correction after orthology clustering, including a new HMM-HMM reciprocal best hit test for orthogroup completion (see, e.g., Supplementary file 1—supplementary table 7). These additional steps (together with careful sampling) distinguish our pipeline from the original OrthoMCL and from other orthology pipelines and influence the interpretation of cluster/orthogroup origin. The additional steps are computationally demanding and require, at present, extensive manual processing. We performed them, as proof-of-principle, for the 157 orthogroups with assumed bilaterian ancestry, but this approach is not yet scalable. Applying the steps for error correction to all clustered outputs for reference proteomes from a public database is therefore not feasible."I am a bit puzzled reading this response. If I understand the answer correctly then the authors state that they perform a semi-automated curation of orthologous groups predicted by orthoMCL, which is absolutely fair. In their manuscript, however, the authors advertise their method as "A unique orthology pipeline for the identification of bilaterian-specific genes", which creates the (untested) impression that the pipeline outperforms existing ortholog search tools. I see two possibilities. Either, the authors present their method as a curation step without the claim of having developed a new ortholog search pipeline. Or, alternatively, they perform a comprehensive benchmark comparing it to existing and state of the art “orthology pipelines”. As a last aspect to consider, part of the QfO benchmark analyses are independent of the QfO reference proteomes. For example, tree discordance tests can be, in principle performed with any underlying set of proteomes.To sum it up, my main concern is not necessarily the method itself, except the orthologous group completion step using hmm-hmm searches, of course. Rather, I am very much worried about the fact that the authors generate the impression of having developed a new pipeline for orthologous group compilation, which apparently performs better than existing software.

The reviewer criticizes that we "generate the impression of having developed a new pipeline for orthology prediction that performs better than existing software" and/or that we "claim of having developed a new ortholog search pipeline". As it has never been our intention to mislead the reader with respect to the nature of our orthology pipeline, we carefully examined the manuscript for the presence of equivocal text passages. We therefore changed the wording of the sentence: from "improved its performance" (the performance of OrthoMCL) to "improved its scalability". We further re-organised the first paragraph of the original version of the Discussion: We split it into several paragraphs with more informative headings (and additions shown in red) and included an entirely new paragraph that addresses various problems mentioned by the reviewer (see also below). In this process, we changed the old header "A unique orthology pipeline for the identification of bilaterian-specific genes" because it also might have fueled misunderstandings. The corresponding new header is "An R-based OrthoMCL pipeline for processing large datasets".

In her/his first comment, reviewer 2 also addresses the high false-positive rate of OrthoMCL and the possibility of tree discordance tests. In the following, we respond to these concerns in detail:

Mentioning the paper by Chen et al., 2007, the reviewer is worried that OrthoMCL's "very high false positive rate of 16%" might compromise our results. Although this rate is indeed a result of the Chen paper, it is important to consider its context: The Chen paper assessed the performance of 10 different orthology detection methods using a -- by today's standard -- small dataset of "27,562 protein sequences from six eukaryotic genomes", which span a large evolutionary interval of 1.3 billion years ( = divergence time between Arabidopsis and *Drosophila* according to timetree.org). When interpreting the Chen et al. orthology clustering results (performance, FP and FN rate etc.), one has to keep in mind that they have been obtained with this comparatively modest dataset. Importantly, we made an effort to compile a more comprehensive and representative dataset (273 opisthokont species with members of 21 metazoan phyla) because uneven phylogenetic coverage (as in Chen et al.) has been identified as a major confounding factor for orthology prediction (p2 of manuscript; Trachana et al., 2011). As the performance of OrthoMCL is dependent on the dataset (and the settings, e.g. of the MCL inflation parameter), and not an inherent property of the algorithm, performance measures of OrthoMCL in our much more comprehensive dataset will be different from the results obtained in the Chen et al. comparison. In fact, OrthoMCL was -- despite the high FP rate -- the best performing algorithm of the three methods generating orthologous clusters in the Chen comparison. To underscore the dependency of orthology prediction on the dataset, we now mention in the Discussion the high false-positive rate of OrthoMCL and the benefits of a comprehensive dataset.

The potentially high false-positive rate of OrthoMCL further indicates that orthogroups might be inferred as bilaterian-specific, but are in fact not, and only appear so because false positive sequences were erroneously assigned to them (discussed in Appendix, error type 3). According to our filtering strategy, this can happen only if two conditions are met: First, the false positive sequences need to belong to a different lineage than the "truly" assigned sequences (e.g., to Ecdysozoa while the other sequences are from Deuterostomia) to induce a change in the orthogroup ancestor to Bilateria. Second, four or more sequences of an ancestor-changing lineage need to be added erroneously (e.g., four ecdysozoans added to an otherwise mammalian-specific orthogroup). While our filtering conditions reject orthogroups with less than four false-positive sequences, we cannot currently detect false-positive bilaterian-specific orthogroups generated by the addition of four or more ancestor-changing sequences. However, this error will only affect a minority of small bilaterian-specific orthogroups with uneven representation. Detailed phylogenetic analysis as well as improved taxon sampling would be necessary to discover such false-positive assignments. To complete error discussion, we included the described effect of false positives in the revised version of the manuscript (Appendix).

The reviewer states that "tree discordance tests can be, in principle performed with any underlying set of proteomes".

We are not sure whether this statement is a request to perform discordance tests. According to Altenhoff et al., 2016, a "species tree discordance test exploits the [orthology] relationship by assessing the accuracy of orthologs in terms of the accuracy of the species tree that can be reconstructed from them". Although it would support orthology if the gene tree of orthologous sequences matches the species tree, the reverse conclusion is not true: We can not necessarily infer non-orthology if species tree and gene tree do not match because of various problems in phylogenetic tree reconstruction (see, for example, Aguileta et al., 2008). Even in their own comparison of orthology pipelines, Altenhoff et al. came to the conclusion that "there is no obvious systematic difference in performance between […] methods relying on species tree […] and methods that do not". We therefore do not see that including discordance tests would add value, or insight, to the conclusions of our article. However, we now mention gene tree-species tree discordance in a new paragraph of the Discussion.

2) My second concern was the limited sensitivity of Blast in ortholog searches. I very much appreciate that the authors have considered this now in the Discussion. In my opinion, some aspects require further attention. Moyers and Zhang, as well as Jain et al. have shown that the sensitivity of Blast can become limiting even when searching for orthologs in the same kingdom. I think it would be a good idea to openly discuss the issue of limited sensitivity ignoring the fact that you “only” search within the same kingdom. It is then straightforward to propose extensive taxon sampling together with the very sensitive hmm-hmm comparisons as a way out of this problem. The shift to a different search method comes then at the cost of a substantially decreased specificity in the ortholog assignment (see my point above). The authors can then explain why they think that an elevated rate of false positive either do not harm, or how they get rid of them in a downstream curation.

As response to the reviewer’s suggestion, we replaced the corresponding paragraph (starting with "Low protein traceability, the failure to detect…") of the Discussion by a new paragraph. We now generally acknowledge the possibility of homology detection failure and describe the benefit of HMM-HMM comparisons for minimizing errors caused by the low traceability of our candidate genes.

3) The authors write in their response "Our primary intention was to generate molecular phylogenies as a means of validating orthogroup clustering, not for inferring specific evolutionary scenarios directly from the gene trees." I cannot follow this argumentation, and I am convinced that I misinterpret the answer. Phylogenetic trees are the basis for deriving evolutionary scenarios on the sequence level, and I cannot see how this cannot be the case. In this context, I regret to say that I find the sentence "The relationship between orthology clusters is imposed by the method and does not necessarily reflect the timing of evolutionary events" impossible to understand. A phylogenetic tree that does not reflect the timing of evolutionary events is either wrong or unrooted. If the latter is the case, then there has to exist the possibility to place the root such that the evolutionary scenario emerges that is proposed in the text. If this is not the case, then again either the tree or the proposed scenario is wrong.Concerning the precise example of FoxH1: Given the tree shown in Figure 5—figure supplement 3, there is no way that FoxH1 is a bilaterian invention. The FoxH1 lineage split from its sister group (either FoxQ1 or FoxD) via a gene duplication prior to the diversification of the bilateria. If no FoxH1 sequences outside the Bilateria were found, then they were either lost-which still does not make FoxH1 a bilaterian invention-or the taxon sampling was not sufficient, and non-Bilaterian FoxH1 sequences await their detection. Alternatively, the tree is wrong, but then it should not be shown. I consider this a major issue.

The reviewer is concerned that "either the tree or the proposed scenario is wrong". We agree that this statement is correct, if evolution proceeded with a strictly constant molecular clock. Unfortunately (for tree reconstruction), this is not the case. Hence, the structure of inferred evolutionary/phylogenetic trees need not always reflect the historically true branching order. In particular, heterotachy of evolutionary rates has been widely documented to accompany, and to last after, events of gene duplication. The cases which we discuss in our manuscript can all be explained by this phenomenon. We made an effort to clarify our explanations in the text by adding a new paragraph to the Discussion ("Challenges in reconciling orthogroups and phylogenetic trees").

We further removed all instances of the unfortunate/unreasonable statement "The relationship between orthology clusters is imposed by the method and does not necessarily reflect the timing of evolutionary events", but we kept the original trees according to the considerations above and the changes in the Discussion.

4) As a response to my concern about missing topology tests, the authors have recomputed the trees and have added approximate SH-LRT support values. They write in their answer "Although the new phylogenies fully support the conclusions derived from the original analysis using RAxML, we acknowledge that some trees show high gene/species topology discordance within a given protein clade(orthogroup). We accept that this is, in part, the consequence of including mixed data types, such as the shorter ORF sequences, and sequence sets not optimized for phylogeny because they are often derived directly from the orthogroups obtained in our clustering." I do not get the point about why it is a problem of using sequences directly derived from orthogroups. Is this again an issue of missing specificity in the ortholog assignments?

Perhaps our previous answer led to misunderstandings. The branching order of phylogenetic trees can be affected by the inclusion of problematic sequences, little phylogenetic information, inappropriate evolutionary models, sampling depth, or heterotachy, to name a few. The presence of (often) rather short ORF-derived sequences is certainly not optimal (in comparison to full length sequences) for phylogenetic analysis and may reinforce gene-tree species-tree discordance. We do not think that such discordance is caused by missing specificity in the orthogroup assignment. If orthogroup assignments were unspecific, we should see low support in the nodes defining an orthogroup/clade. This is not the case for the phylogenetic trees shown in our article. We are thus confident that the sequences assigned to a specific orthogroup do belong to this orthogroup. In contrast, the branching order of sequences within orthogroups does not necessarily match the expected species relationships (gene tree/species tree discordance) for the various reasons outlined above. As expected, support values for branching events within orthogroups are therefore often low. We now discuss these considerations in the new paragraph of the Discussion ("Challenges in reconciling orthogroups and phylogenetic trees").

[Editors' note: further revisions were suggested prior to acceptance, as described below.]

[…] We see two ways out of this dilemma:One is for you to perform additional analyses, which will take time. This would likely have to include the results of the ortholog searches in a well-defined set of species, such that presence/absence of orthologs can be interpreted along a (known species) tree. For example, at the moment, it is unclear why the tree of Nodal features about 30 vertebrate sequences, that of FoxH1 only about 20 sequences, and that of EOMES only 2. Although these numbers are not relevant when it comes to the age of the genes, they tell us something about the precision and rigor of the analysis.The other possibility is that the authors could modify the manuscript to avoid the term “innovation” where it is not backed up by the data. If the tree argues for an evolutionarily old gene, which nowadays – and according to the analyses of the authors – is confined to the bilateria, why not simply call it a “bilaterian-specific” gene, leaving it for the moment open whether it was indeed an “innovation” or a selective retention of an older gene. In essence, the authors would then argue with their phylogenetic profiles, which seem to be comprehensive, and not with trees, which they indicate in the response might be wrong anyways. If the authors then wish, they could speculate that the gene is indeed a bilateral innovation, but it has to be clear that this is a speculation that would require more thorough analyses to be tested.

We very much appreciate that you mentioned the complexity of our study and the effort needed to perform additional analyses, such as relative rate tests on the branches of the phylogenies shown in Figure 5—figure supplement 3, Figure 5—figure supplement 4, or Figure 7—figure supplement 1. While we agree that such analyses may yield additional insight, we think that they are beyond the scope of the present manuscript and should be deferred to a future study. We therefore followed your second suggestion and main comment: to modify the manuscript and avoid the term "innovation" where it is not clearly supported by the data.

Throughout the manuscript, we therefore changed the text when our conclusions (bilaterian origin/innovation) were in conflict with phylogenetic trees. As suggested, we now describe the respective genes/features as bilaterian-specific or as having a bilaterian-wide distribution, leaving their evolutionary origin open.

We now also acknowledge in the discussion the existence of gene loss as an important evolutionary mechanism and cite appropriate studies, as requested by the reviewer. Despite the increasing recognition of gene loss and the studies mentioned by the reviewer, there is, to our knowledge, still no support in the literature for a scenario comparable to the one some of our trees imply: the parallel loss of several genes in several entire sister phyla (such as the loss of FoxH1 in cnidarians, poriferans, placozoans, and ctenophores).

To seek further support for our interpretation of trees, a rate acceleration of the FoxH branch after duplication (and therefore origin in the bilaterian ancestor), we analysed in a preliminary way the CDSs of human FoxD/FoxH/FoxQ sequences using Tajima's 1- and 2-parameter relative rate tests (Tajima, Genetics, 1993 Oct;135(2):599-607). Although the results in this special case confirmed our claim, a thorough analysis on the basis of several genes from several organisms should be the focus of a different study.